# Tau deposition patterns are associated with functional connectivity in primary tauopathies

Nicolai Franzmeier [1,36✉], Matthias Brendel [2,3,36], Leonie Beyer [2], Luna Slemann[2], Gabor G. Kovacs [4,5,6], Thomas Arzberger[3,7,8,9], Carolin Kurz[7,8], Gesine Respondek[7,10,11], Milica J. Lukic[7,12], Davina Biel[1], Anna Rubinski[1], Lukas Frontzkowski[1], Selina Hummel[2], Andre Müller[13], Anika Finze [2], Carla Palleis[3,7,14], Emanuel Joseph[2], Endy Weidinger[14], Sabrina Katzdobler [14], Mengmeng Song[2], Gloria Biechele[2], Maike Kern[2], Maximilian Scheifele[2], Boris-Stephan Rauchmann[15], Robert Perneczky [3,7,8,16], Michael Rullman[17], Marianne Patt[17], Andreas Schildan [17], Henryk Barthel[17], Osama Sabri[17], Jost J. Rumpf[2,18], Matthias L. Schroeter[2,18], Joseph Classen[19], Victor Villemagne[20,21,22], John Seibyl[23,24], Andrew W. Stephens [13], Edward B. Lee[4], David G. Coughlin [25,26], Armin Giese[9], Murray Grossman[25,27], Corey T. McMillan[25,27], Ellen Gelpi[28,29], Laura Molina-Porcel [28,29], Yaroslau Compta[30], John C. van Swieten[31], Laura Donker Laat[32], Claire Troakes [33], Safa Al-Sarraj[33], John L. Robinson[4], Sharon X. Xie[34], David J. Irwin [10,27], Sigrun Roeber[9], Jochen Herms [7], Mikael Simons[7], Peter Bartenstein[2], Virginia M. Lee[4], John Q. Trojanowski [4], Johannes Levin[3,7,14], Günter Höglinger [7,35,37] & Michael Ewers [1,7,37]

Tau pathology is the main driver of neuronal dysfunction in 4-repeat tauopathies, including cortico-basal degeneration and progressive supranuclear palsy. Tau is assumed to spread prion-like across connected neurons, but the mechanisms of tau propagation are largely elusive in 4-repeat tauopathies, characterized not only by neuronal but also by astroglial and oligodendroglial tau accumulation. Here, we assess whether connectivity is associated with 4R-tau deposition patterns by combining resting-state fMRI connectomics with both 2nd generation [18]F-PI-2620 tau-PET in 46 patients with clinically diagnosed 4-repeat tauopathies and post-mortem cell-type-specific regional tau assessments from two independent progressive supranuclear palsy patient samples ($n = 97$ and $n = 96$). We find that inter-regional connectivity is associated with higher inter-regional correlation of both tau-PET and post-mortem tau levels in 4-repeat tauopathies. In regional cell-type specific post-mortem tau assessments, this association is stronger for neuronal than for astroglial or oligodendroglial tau, suggesting that connectivity is primarily associated with neuronal tau accumulation. Using tau-PET we find further that patient-level tau patterns are associated with the connectivity of subcortical tau epicenters. Together, the current study provides combined in vivo tau-PET and histopathological evidence that brain connectivity is associated with tau deposition patterns in 4-repeat tauopathies.

A full list of author affiliations appears at the end of the paper.

Progressive supranuclear palsy (PSP) and cortico-basal degeneration (CBD) are primary 4-repeat (4 R) tauopathies characterized by glial and neuronal 4R-tau inclusions, manifesting as progressive movement and cognitive disorders[1–4]. In PSP and CBD, 4R-tau pathology accumulates initially in the brainstem and subcortex, with subsequent cortical manifestation at more advanced disease stages[5, 6].

Postmortem studies have reported a strong clinico-pathological correlation between 4R-tau deposition patterns and clinical phenotype: PSP most commonly presents as Richardson's syndrome (PSP-RS, i.e., a combination of postural instability and ocular motor deficits), associated with brainstem and subcortical tau followed by progressive tau accumulation in parietal and motor cortices at later disease stages[7, 8]. Further variant predominance types include, among others, PSP with speech/language disorder with left inferior frontal tau aggregation or PSP with cognitive/behavioral presentation with fronto-temporal tau pathology, suggesting that clinical variability is driven by spatially heterogeneous tau patterns[8–10]. In accord with this concept, CBD patients often present with mixed cortical/movement disorders termed cortico-basal syndrome (CBS), associated with tau accumulation in the motor cortex, brainstem, subthalamic nucleus, and striatum, yet clinical variants present as progressive aphasia, frontal-behavioral syndrome or Richardson's syndrome[11, 12]. Since 4 R tau deposition patterns in PSP and CBD are considered key determinants of disease phenotype and progression, a detailed understanding of the mechanisms that facilitate tau spreading is of pivotal clinical interest.

There is converging preclinical evidence suggesting cell-to-cell tau pathology transmission across functionally interconnected neurons:[13] First, pathological tau strains obtained from patients with primary or secondary tauopathies have been shown to induce template-based misfolding of physiological tau, suggesting prion-like tau propagation[14, 15]. Second, injection of pathological tau seeds in mouse brains triggers tau spread to regions anatomically connected to the injection site rather than tau diffusion to spatially adjacent regions[15–19]. Third, synapses and neuronal activity are considered to be key drivers of tau spreading[20, 21], where opto-genetically enhanced activity of tau harboring neurons is associated with accelerated trans-synaptic tau spreading in vitro and in vivo[22]. Collectively, these preclinical findings provide strong support for neuronal connectivity as a key route for tau spreading in tauopathies. However, it remains unclear whether tau pathology progresses preferentially between connected brain regions in human 4 R tauopathies. This spreading mechanism needs to be specifically questioned in 4 R tauopathies since tau is exclusively present not only in neurons but also in astroglia and oligodendroglia[8].

Here, we combined in vivo tau-PET and postmortem tau assessments from 4 R tauopathy patients with fMRI-based connectivity assessments obtained in a healthy reference sample. Using [18]F-PI-2620 PET for tau imaging in clinically diagnosed CBS ($n = 24$) and PSP-RS patients ($n = 22$), we tested if (i) functionally connected brain regions show correlated [18]F-PI-2620 PET levels and whether (ii) brain-wide tau-PET uptake patterns are associated with the functional connectivity pattern of subcortical epicenters with highest tau pathology. In CBS, which is typically associated with more widespread cortical tau[23], we further tested whether subthreshold cortical Aβ levels may enhance the spread of tau-PET from subcortical tau epicenters to the cortex. This analysis was motivated by previous reports showing clinical and pathophysiological overlap between CBS and AD[24], where a substantial number of clinical CBS cases had co-occurring Aβ pathology[23, 25], i.e., a key driver of cortical tau spreading in AD[26]. In a key validation step, we translated the in vivo tau-PET analyses to two independent postmortem samples (Munich-European consortium/collection): $n = 97$, University of Pennsylvania [UPENN]: $n = 96$) with confirmed PSP pathology and gold-standard regional histopathological 4 R tau assessments and further performed in vitro binding assays and autoradiographic analyses that support PI-2620 binding to 4 R tau. Inclusion of this postmortem sample allowed cell-type-specific stratification of tau pathology (i.e., neuronal, astroglial, and oligodendroglial tau) to specifically test the hypothesis that inter-regional connectivity is associated with the deposition of neuronal tau pathology. By combining cell-type-specific postmortem tau assessments with atlas-based functional connectivity data, we tested the hypothesis that higher functional connectivity between postmortem sampled brain regions is associated with more correlated tau levels, and that this association is primarily driven by neuronal tau. Our results confirm that functional connectivity is associated with subcortical and cortical tau-PET patterns in PSP and CBS patients and that the association between functional connectivity and histopathologically assessed tau is strongest for neuronal tau levels.

## Results
We obtained tau-PET in 46 patients with clinically suspected 4 R tauopathies, including 24 CBS patients, 22 PSP-RS patients, and 15 cognitively normal controls (CN). All subjects underwent 3 T structural MRI and [18]F-PI-2620 PET imaging[27]. Aβ-positive CBS patients (as determined by cerebrospinal fluid Aβ levels or amyloid-PET, which was available for all CBS cases) were considered to have underlying 3 R/4 R tau AD pathology and were therefore excluded from the current study[28]. Sample characteristics are presented in Table 1. Group-average [18]F-PI-2620 SUVR maps intensity normalized to the inferior cerebellar gray matter are shown in Fig. 1 for CN (Fig. 1A), PSP-RS (Fig. 1B) and CBS (Fig. 1C). For neuropathological validation of the tau-PET data, we included regional postmortem tau assessments from two independent samples (Munich-European consortium/collection sample, $n = 97$; UPENN sample, $n = 96$) with histopathologically confirmed PSP, defined as the presence of neurofibrillary tau tangles (NFT) in the subthalamic nucleus, substantia nigra, and globus pallidus together with tufted astrocytes in the striatum and frontal cortex[3, 8]. In addition, we included postmortem data from 16 patients with histopathologically confirmed PSP for autoradiographic assessment of PI-2620 binding to tau pathology. For functional connectivity, we included resting-state fMRI data of 69 cognitively normal subjects from the Alzheimer's disease neuroimaging initiative (ADNI, age = $67.89 \pm 5.88$, sex[f/m] = 39/30) cohort without evidence of amyloid or tau pathology as indicated by [18]F-florbetapir amyloid-PET and [18]F-flortaucipir tau-PET[29, 30]. Age and sex in the ADNI resting-state fMRI sample were not statistically different from age and sex distributions in the PSP-RS or CBS groups ($p > 0.05$).

**Elevated tau-PET binding in PSP-RS/CBS.** First, we assessed [18]F-PI-2620 PET binding in CBS and PSP-RS patients vs. CN, using voxel-wise ANCOVAs, controlling for age, sex, and study site (alpha-threshold = 0.005, cluster-extent threshold > 100 voxels). Compared to CN, PSP-RS patients showed higher subcortical and inferior frontal tau-PET binding (Fig. 1D), congruent with previous work in a partly overlapping sample[27]. In CBS[31], we not only found significantly elevated subcortical, brainstem, and inferior frontal[18]F-PI-2620 PET binding but also more widespread occipital, frontal, and parietal elevations of [18]F-PI-2620 PET binding (Fig. 1E). When comparing [18]F-PI-2620 PET between CBS and PSP (Fig. 1F), we found higher cortical [18]F-PI-2620 PET binding in CBS in the midbrain, entorhinal and lateral temporal cortex, motor cortex, anterior cingulate, frontal eye

**Table 1 Subject demographics.**

**Tau-PET sample (N = 61)**

| | Controls (n = 15) | PSP-RS (n = 22) | CBS (n = 24) | P |
|---|---|---|---|---|
| Age | 62.67 ± 8.97[b] | 72.07 ± 6.40[a] | 67.71 ± 8.46 | 0.001 |
| Sex (f/m) | 9/6 | 10/18 | 14/10 | 0.175 |
| Disease duration (months) | NA | 39.48 ± 25.01 | 28.86 ± 21.48 | 0.143 |
| PSP Rating Scale | NA | 36.21 ± 15.12[d] | 24.95 ± 11.85[e] | 0.011 |
| SEADL | NA | 55.5 ± 20.89[f] | 64.36 ± 20.89[e] | 0.162 |
| MoCA | NA | 22.73 ± 4.10[g] | 23.18 ± 5.39[e] | 0.807 |

*MoCA* Montreal Cognitive Assessment Battery, *SEADL* Schwab and England Activities of Daily Living.
[a]Significantly different from controls ($p < 0.05$).
[b]Significantly different from PSP-RS ($p < 0.05$).
[c]Available for 19/22 PSP-RS subjects.
[d]Available vor 22/24 CBS subjects.
[e]Available for 20/22 PSP-RS subjects.
[f]Available for 11/22 PSP-RS subjects.

fields, and parietal cortex. In line with histopathological studies[3], these analyses suggest elevated tau in patients with suspected 4 R tauopathies vs. CN, particularly in the subcortex, with more widespread cortical tau in CBS vs. PSP-RS[23].

**PI2620 signal reflects 4 R tau pathology**. To validate whether PI-2620 has the ability to detect 4 R tau, we performed an in vitro competition assay to compare the affinity of PI-2620 to 4 R tau fibrils vs. the 1st-generation tau-PET tracer Flortaucipir. As shown in Fig. 2A, we found higher IC50 values of PI-2620 to 4 R tau fibrils (IC50 = 2.7 nM) compared to Flortaucipir (IC50 = 18.4 nM), suggesting that PI-2620 shows higher in vitro affinity to 4 R tau. In addition, we performed combined PI-2620 autoradiography and AT8 immunohistochemistry in 233 postmortem probes derived from 16 histopathologically confirmed PSP cases across three brain regions (i.e., frontal cortex: $n = 105$, pallidum: $n = 56$; putamen: $n = 72$). AT8 staining was assessed semiquantitatively by expert neuropathologists (i.e., low = +, medium = ++, high = +++) and autoradiography was quantified as the ratio of PI-2620 signal in the target tissue (i.e., frontal cortex, pallidum, and putamen) divided by PI-2620 signal in AT8-negative white matter. Using spearman correlation, we found a positive association between semiquantitative AT8 assessments and PI-2620 autoradiographic signal (Frontal cortex: r = 0.44, $p < 0.001$; Pallidum: r = 0.47, $p < 0.001$; Putamen: r = 0.4, $p < 0.001$), as well as significant differences in PI-2620 autoradiography signals between semiquantitatively assessed AT8 staining groups using ANOVAs (Frontal cortex: F = 10.42, $p < 0.001$; Putamen: F = 12.08, $p < 0.001$; Pallidum: F = 8.36, $p < 0.001$, Fig. 2B). Together, these results suggest that PI-2620 can detect 4 R tau pathology. Example images for the autoradiographic PI-2620 signal vs. AT8 staining are shown in Fig. 2C for the Frontal cortex and Fig. 2D for the basal ganglia.

**Functional connectivity is associated with correlated tau-PET binding in CBS/PSP-RS**. Next, we tested the major hypothesis whether inter-regional connectivity is associated with inter-regionally correlated tau accumulation in CBS or PSP-RS. To this end, we parcellated patient-specific 18F-PI-2620 PET images into 232 regions-of-interests (ROIs) by combining non-overlapping functional MRI-based parcellations of the cortex (200 ROIs, Fig. 1G)[32] and subcortex (32 ROIs, Fig. 1H)[33]. Primary analyses focused on the subcortex (32 ROIs, Fig. 1G), which showed consistently elevated 18F-PI-2620 PET binding in CBS and PSP-RS (see Fig. 1E, F). For secondary analyses we also included the neocortex (i.e., 32 subcortical plus 200 cortical ROIs, Fig. 1G),

which was less consistently affected by elevated 18F-PI-2620 PET binding in PSP-RS/CBS (see Fig. 1E, F).

We determined the covariance in inter-regional 18F-PI-2620 PET binding within CBS and PSP-RS groups, defined as partial correlation of ROI-based 18F-PI-2620 PET SUVRs, accounting for age, sex, and imaging site as confounds (methods illustrated in Fig. 3A, for subcortical tau covariance matrices see Fig. 3B for PSP-RS and Fig. 3C for CBS). For functional connectivity, we applied the same 232 ROI-based parcellation to preprocessed resting-state fMRI data of 69 cognitively normal ADNI subjects. Based on this healthy control sample, we determined a group-average functional connectivity template (see Fig. 3D). Using linear regression, we next assessed whether higher inter-regional functional connectivity was associated with higher inter-regional covariance of 18F-PI-2620 PET binding, controlling for between-ROI Euclidean distance to ensure that associations were not driven by spatial proximity. For our primary analyses on the subcortex (i.e., where tau-PET was significantly elevated in PSP-RS and CBS compared to controls), we found the expected associations between higher functional connectivity and higher 18F-PI-2620 PET covariance in PSP-RS (β = 0.616, $p < 0.001$, Fig. 4A) and CBS (β = 0.561, $p < 0.001$, Fig. 4B). To test the robustness of these findings, we recomputed these models 1000 times generating for each trial a new connectivity null-model (i.e., shuffled connectivity matrix with preserved weight- and degree distribution), yielding a distribution of null-model derived β-values (Fig. 4A, B, beeswarm panels). Comparing the actual β-values estimated using the observed true connectivity matrix with the null-distributions using an exact test (i.e., determine the probability of null-distribution derived β-values surpassing the true β-value), yielded p-values <0.001 for CBS and PSP-RS. Together, these analyses support the view that functionally connected subcortical regions show covarying tau-PET levels in CBS/PSP-RS. We obtained consistent results when extending these analyses to the whole brain (Fig. 1G, H), where higher connectivity was again associated with higher 18F-PI-2620 PET covariance in PSP-RS (Fig. 4, β = 0.450, $p < 0.001$) and CBS (Fig. 4D, β = 0.402, $p < 0.001$), controlling for inter-regional Euclidean distance. Exact tests using shuffled connectivity data yielded consistent results ($p < 0.001$, Fig. 4C, D, beeswarm panels). Repeating the above-described analyses using partial-volume corrected tau-PET data or omitting Euclidean Distance as a covariate yielded consistent results (Supplementary Fig. 1, Supplementary Tables 1, 3). We next tested whether the association between connectivity and covariance in tau-PET was consistent across subcortico-cortical and cortico-cortical connections. Again, associations between connectivity and 18F-PI-2620 PET covariance were found for cortico-cortical (PSP-RS: β = 0.397, $p < 0.001$; CBS: β = 0.397, $p < 0.001$) and subcortico-cortical

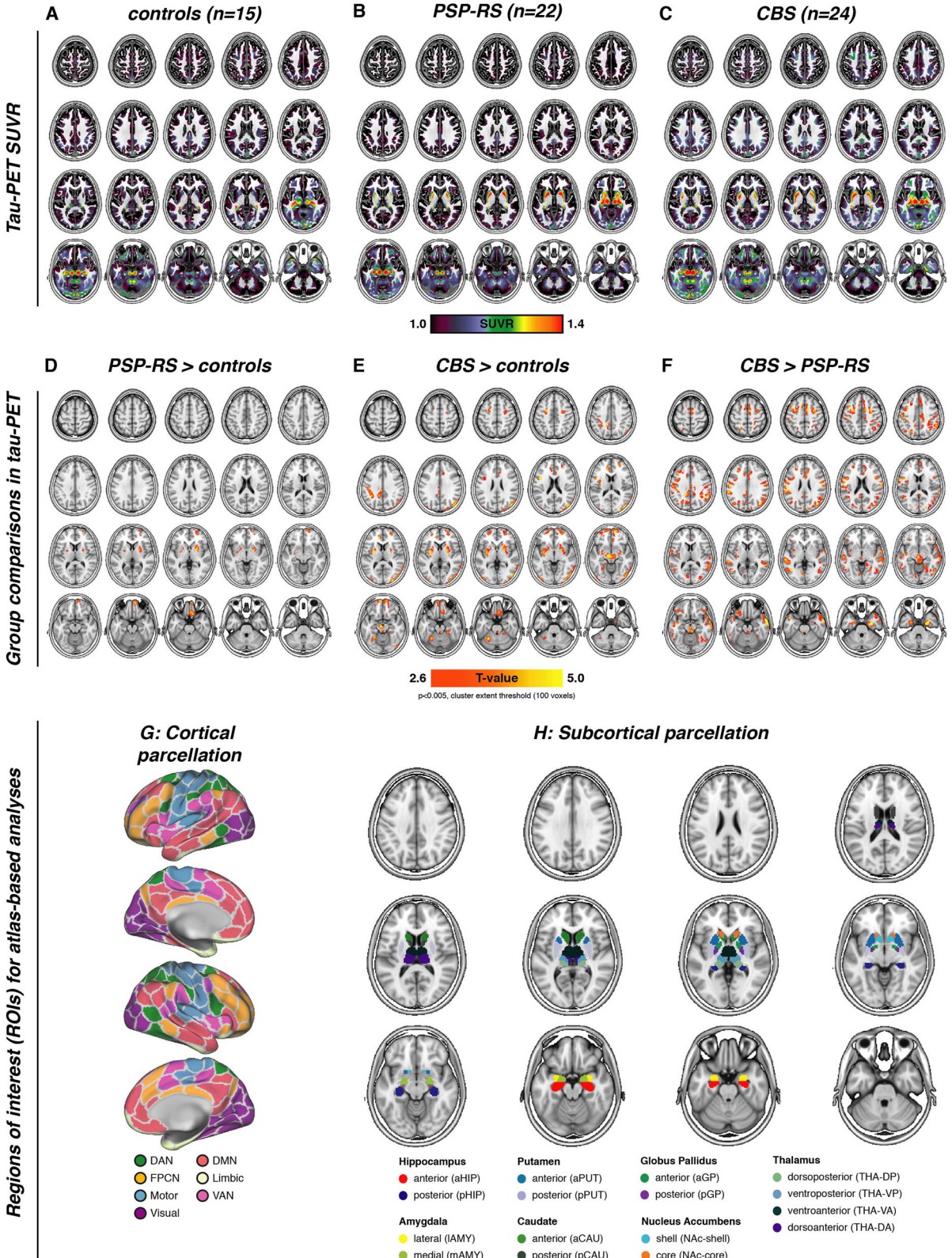

**Fig. 1 PI-2620 tau-PET uptake patterns in controls/patients and brain parcellation scheme.** Group-average maps of tau-PET SUVRs (i.e., intensity normalized to the inferior cerebellar gray) for controls (**A**), PSP-RS (**B**), and CBS patients (**C**). Voxel-wise group comparisons were conducted to compare tau-PET SUVRs between PSP-RS vs. controls (**D**), CBS vs. controls (**E**), and CBS vs. PSP-RS (**F**), at a voxel threshold of *p* < 0.005 with a cluster size of at least 100 spatially contiguous voxels. Illustration of the 200 ROI cortical (**G**) and 32 ROI subcortical (**H**) brain atlases that were used for all tau-PET vs. connectivity analyses.

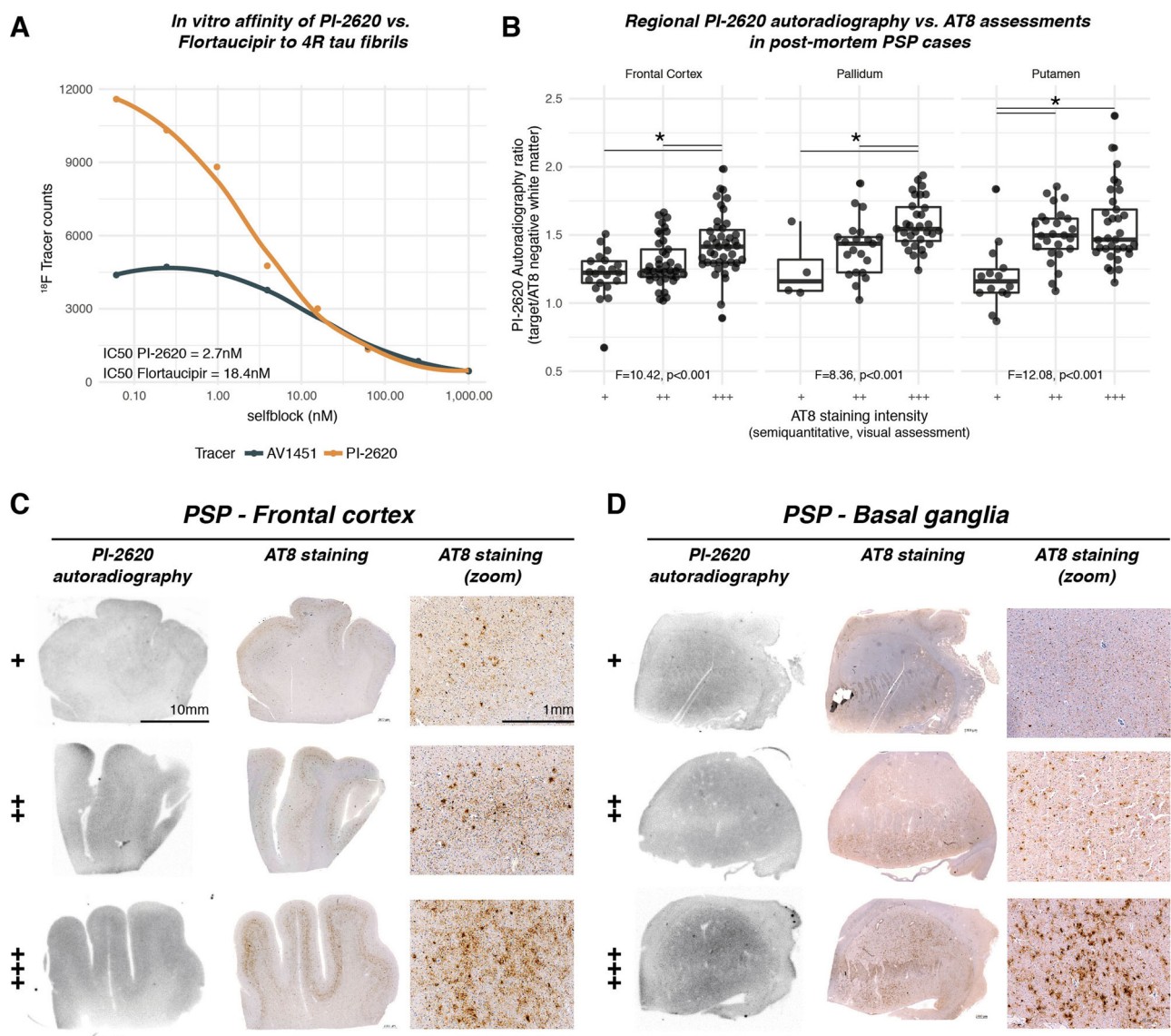

**Fig. 2 Autoradiographic assessment of PI-2620 binding in PSP patients.** In vitro competition assay, showing stronger affinity of PI-2620 to 4R tau fibrils than the first generation tau-PET tracer Flortaucipir (**A**). In addition, we performed autoradiographic assessments in 233 brain samples derived from 16 patients with histopathologically confirmed PSP pathology. Samples were obtained from the frontal cortex ($n = 105$), pallidum ($n = 56$), and putamen ($n = 72$). AT8 staining intensity was judged by visual expert read (low = +, medium = ++, high = +++), autoradiography was quantified as the intensity of the autoradiographic signal in the target tissue divided by signal in AT8-negative white matter. Boxplot illustrating the comparisons between autoradiography signal and AT8 staining intensity are shown in **B**, examples of autoradiography samples and AT8 staining are shown for the frontal cortex in **C**, and the basal ganglia in **D**. Two-sided $p$-values have been determined via ANOVAs. Boxplots are displayed as median (center line) ± interquartile range (box boundaries) with whiskers including observations falling within the 1.5 interquartile range. Source data are provided as a Source Data file.

connections (PSP-RS: $\beta = 0.329$, $p < 0.001$; CBS: $\beta = 0.379$, $p < 0.001$). Together, these findings of correlated tau-PET among functionally connected regions support the idea that connectivity shapes tau deposition patterns in 4 R tauopathies.

**Functional connectivity predicts tau-PET binding in PSP-RS/CBS**. We next determined whether group-average 18F-PI-2620 PET patterns follow the connectivity pattern of subcortical tau epicenters (i.e., 20% of brain regions with highest tau-PET). We found that for epicenter regions with highest 18F-PI-2620 PET binding, higher seed-based functional connectivity was associated with higher tau-PET binding in strongly connected subcortical regions in both PSP-RS (Fig. 5A, $\beta = 0.880$, $p < 0.001$) and CBS (Fig. 5B, $\beta = 0.933$, $p < 0.001$), controlling for between-ROI Euclidean Distance. In turn, for coldspot regions with lowest 18F-PI-2620 PET binding, higher

seed-based functional connectivity was associated with lower subcortical 18F-PI-2620 PET binding in strongly connected regions in PSP-RS (Fig. 5, $\beta = -0.613$, $p < 0.001$) and CBS (Fig. 5D, $\beta = -0.617$, $p < 0.001$), controlling for between-ROI Euclidean distance. This result pattern was reflected in a strong positive correlation between the seed ROIs 18F-PI-2620 PET binding and their functional connectivities' predictive weight (i.e., regression-derived $\beta$-value) on 18F-PI-2620 PET binding in remaining ROIs (PSP-RS: $\beta = 0.929$, $p < 0.001$, Fig. 5E; CBS: $\beta = 0.937$, $p < 0.001$, Fig. 5F). A fully congruent result pattern was detected when extending this approach to the whole brain (see Fig. 5G–L). Consistent results were detected when repeating the analyses using partial-volume corrected data or when omitting Euclidean distance as a covariate (Supplementary Fig. 2; supplementary Tables 1, 3). Together, these findings suggest that regions with high tau-PET levels are primarily

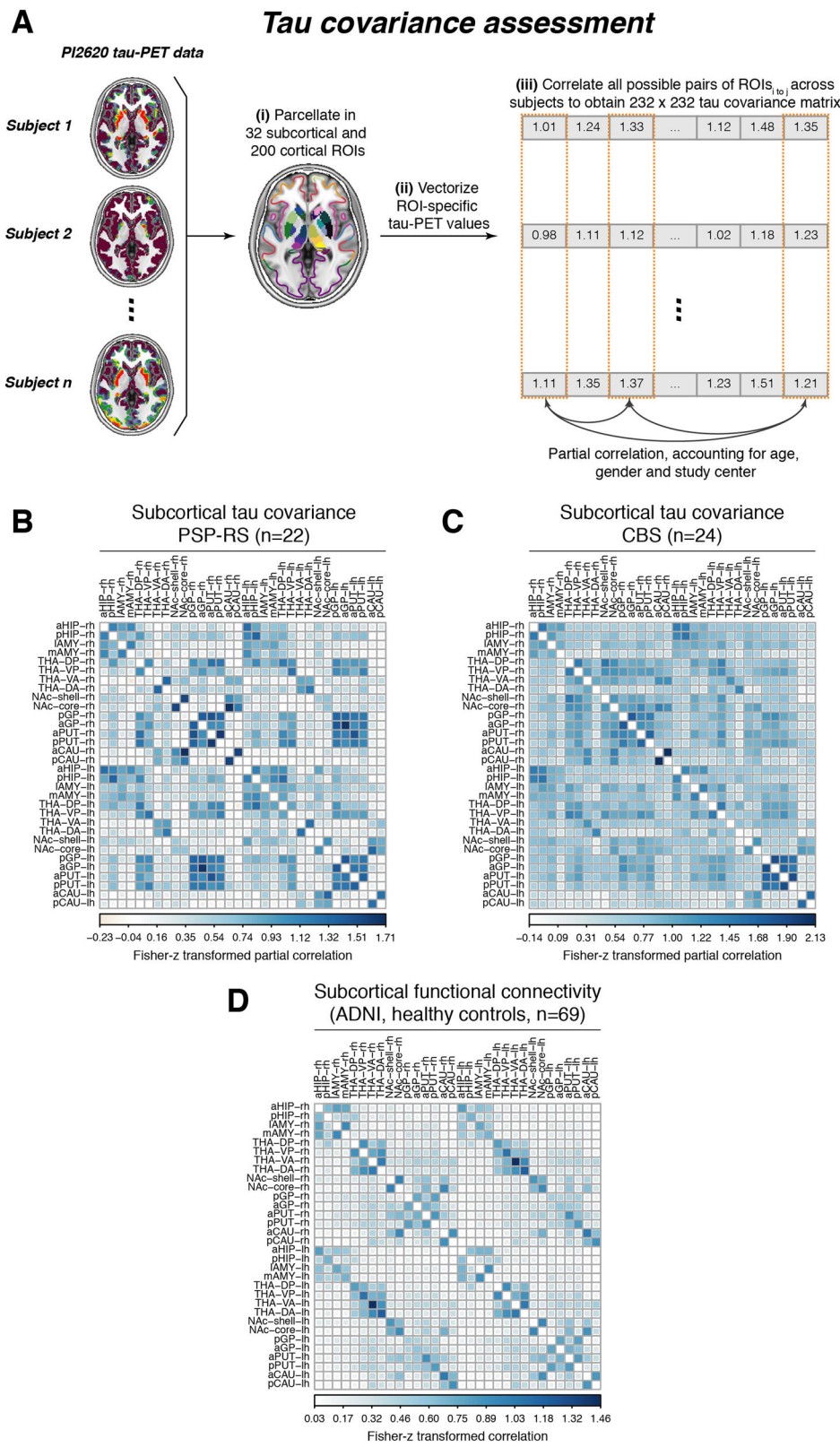

**Fig. 3 Assessment of tau covariance.** Flow-chart illustrating the assessment of tau covariance (**A**). Subject-level tau-PET data were parcellated into 200 cortical and 32 subcortical ROIs (i), mean tau-PET was extracted for each region of interest (ROI), vectorized to 232-element vectors and concatenated across subjects (ii). Fisher-z transformed partial correlations between inter-regional tau-PET SUVRs were determined for each group (i.e., progressive supranuclear palsy—Richardson syndrome [PSP-RS] and cortico-basal syndrome [CBS]), accounting for age, sex, and study site (iii). The resulting tau covariance matrices for the subcortical brain parcellation which was used for primary analyses is shown for PSP-RS (**B**) and CBS (**C**) patients. For the same ROIs, group-average functional connectivity was computed based on resting-state fMRI of 69 cognitively normal, amyloid and tau negative ADNI participants (**D**). Source data are provided as a Source Data file.

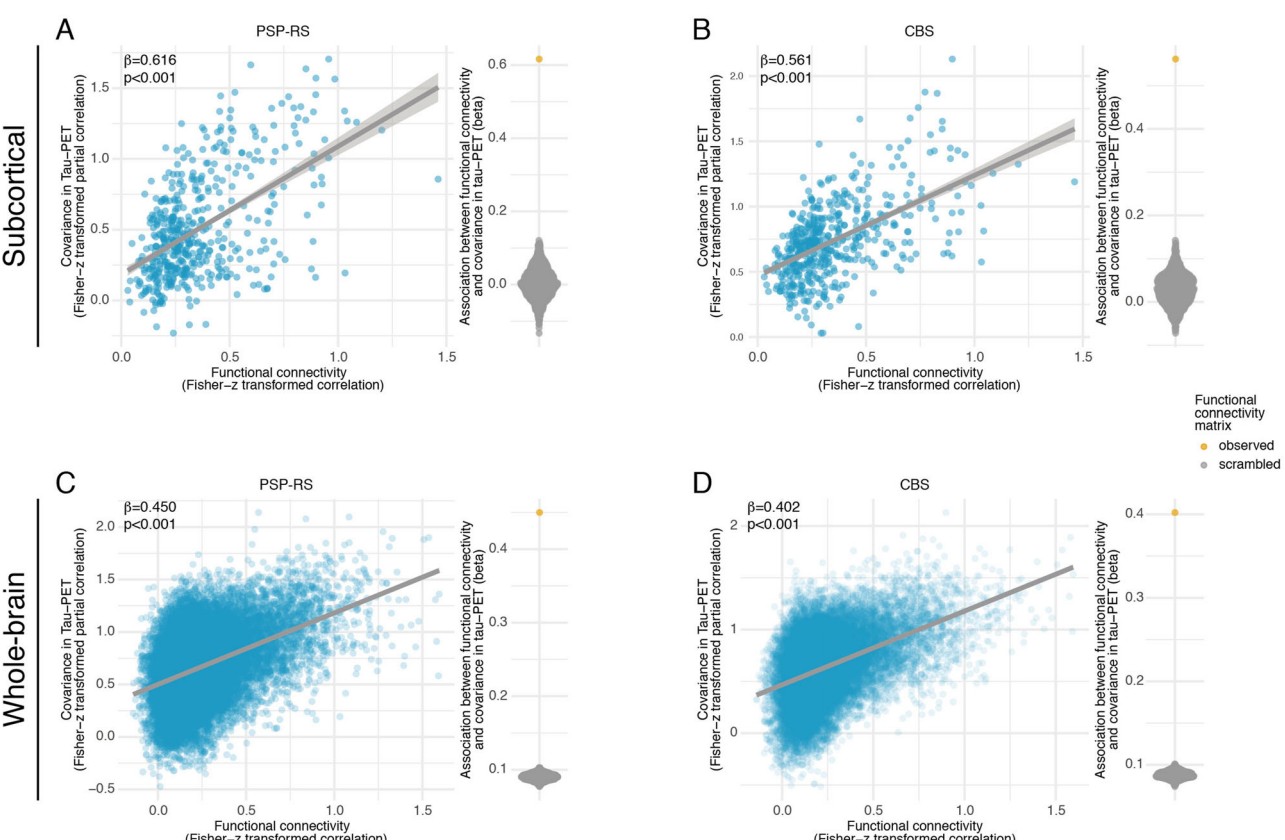

**Fig. 4 Association between connectivity and covariance in tau-PET.** Scatterplots illustrating the association between functional connectivity and covariance in 18-F-PI2620-PET among subcortical regions in progressive supranuclear palsy—Richardson syndrome (PSP-RS, **A**) and cortico-basal syndrome (CBS) groups (**B**), as well as among subcortical and cortical regions for PSP-RS (**C**) and CBS groups (**D**). Standardized beta- and p-values were derived from linear regression controlling for Euclidean distance between regions of interest (ROIs). Beeswarm plots illustrate the distribution of standardized beta-values derived from repeating the analysis 1000 times using scrambled connectomes with preserved weight- and degree distribution (gray points) vs. the beta-value derived from the association with the actual observed connectivity matrix that is illustrated in the scatterplot (yellow point). Two-sided p-values have been determined via linear regression. Linear model fits are indicated together with 95% confidence intervals. Source data are provided as a Source Data file.

interconnected with regions also harboring high tau, whereas regions with low tau are primarily connected to other low tau regions.

**Connectivity of patient-specific tau epicenters predicts individual tau-PET binding in 4 R tauopathy patients.** As shown by the group-level analyses, epicenter regions with highest tau-PET binding were most strongly connected to other regions with high [18]F-PI-2620 PET in PSP-RS (Fig. 5A, G) and CBS (Fig. 5B, H). Next, we extended this analysis to the subject level, i.e., we defined each PSP-RS and CBS patients' tau epicenter as those ~20% of subcortical ROIs with the highest [18]F-PI-2620 PET SUVR[34]. Adopting our previously established approach[34], the remaining non-epicenter subcortical ROIs were grouped for each subject into 4 quartiles depending on the mean connectivity strength to a given subjects' tau epicenter (Q1 = strongest connectivity to the tau epicenter vs. Q4 = weakest connectivity to the tau epicenter) as illustrated in Fig. 6A. We expected a gradient of [18]F-PI-2620 PET decrease from tau epicenters across functionally connected regions (i.e., highest tau in Q1, vs. lowest tau in Q4). Using linear mixed models, we could confirm that connectivity strength to the epicenter was predictive of [18]F-PI-2620 PET binding in connected Q1–Q4 regions for PSP-RS (b-value/standard error [b/SE] = 0.097/0.015, p < 0.001, Fig. 6B) and

CBS (Fig. 6D, b/SE = 0.086/0.013, p < 0.001). As hypothesized, highest [18]F-PI-2620 PET binding was found in Q1, which is most strongly connected to the tau epicenter, whereas tau-PET levels gradually decreased to Q4, which is only weakly connected to the tau epicenter. Linear mixed models were controlled for age, sex, study site, mean Euclidean distance to the tau epicenter as well as random intercept. When extending this analysis to the cortex, we found a similar result pattern, with highest [18]F-PI-2620 PET binding in those cortical regions that were most closely connected to subcortical tau epicenters (i.e., Q1) vs. lowest [18]F-PI-2620 PET binding in those cortical regions that were most weakly connected to subcortical tau epicenters (i.e., Q4) in PSP-RS (Fig. 6C, b/SE = 0.036/0.008, p < 0.001) and CBS (Fig. 6E, b/SE = 0.032/0.005, p < 0.001). A topological frequency mapping of tau epicenters is shown in Fig. 6H for PSP-RS and Fig. 6I for CBS. Consistent results were obtained when using partial-volume corrected tau-PET data or when omitting Euclidean distance as a covariate (supplementary Fig. 3; supplementary Tables 2, 4). Together, these findings support the view that tau deposition patterns follow the connectivity pattern of subcortical tau epicenters in PSP-RS and CBS.

**Subthreshold amyloid levels are associated with neocortical tau deposition in CBS.** In a subset of 22 CBS patients with available

## Functional connectivity associated with tau-PET uptake

**Subcortical**

A — PSP-RS: β=0.880, p<0.001 — Tau-PET SUVR vs Tau epicenter connectivity
C — PSP-RS: β=-0.613, p=0.007 — Tau-PET SUVR vs Tau coldspot connectivity
E — PSP-RS: β=0.929, p<0.001 — Association between Seed FC and Tau-PET in connected regions (beta-value) vs Tau-PET in Seed ROI
B — CBS: β=0.933, p<0.001 — Tau-PET SUVR vs Tau epicenter connectivity
D — CBS: β=-0.617, p=0.006 — Tau-PET SUVR vs Tau coldspot connectivity
F — CBS: β=0.937, p<0.001 — Association between Seed FC and Tau-PET in connected regions (beta-value) vs Tau-PET in Seed ROI

**Whole-brain**

G — PSP-RS: β=0.688, p<0.001 — Tau-PET SUVR vs Tau epicenter connectivity
I — PSP-RS: β=-0.241, p=0.002 — Tau-PET SUVR vs Tau coldspot connectivity
K — PSP-RS: β=0.788, p<0.001 — Association between Seed FC and Tau-PET in connected regions (beta-value) vs Tau-PET in Seed ROI
H — CBS: β=0.689, p<0.001 — Tau-PET SUVR vs Tau epicenter connectivity
J — CBS: β=-0.242, p=0.002 — Tau-PET SUVR vs Tau coldspot connectivity
L — CBS: β=0.761, p<0.001 — Association between Seed FC and Tau-PET in connected regions (beta-value) vs Tau-PET in Seed ROI

Legend: ● DAN ● DMN ● FPCN ○ Limbic ● Motor ● VAN ● Visual ● Subcortical

$^{18}$F-flutemetamol amyloid-PET (i.e., all rated Aβ-negative on expert visual read), we tested the hypothesis that subthreshold Aβ levels are associated with enhanced cortical tau at a given level of subcortical tau deposition. Here, we repeated the subject-level analyses illustrated in Fig. 6D, E, this time including a main effect for high vs. low Aβ as defined via median split of global $^{18}$F-flutemetamol amyloid-PET SUVRs, intensity normalized to

the pons. No difference in $^{18}$F-PI-2620 PET binding was found within subcortical tau epicenters between high and low Aβ groups ($p > 0.05$). Further, the association between epicenter connectivity and $^{18}$F-PI-2620 PET binding in connected subcortical Q1–Q4 ROIs was the same across high/low Aβ groups (main effect of Aβ: b/SE = −0.02/0.051, $p = 0.653$, Fig. 6F), controlling for age, sex, study site, Euclidean distance to the tau epicenter, and random

**Fig. 5 Group-level epicenter connectivity vs. tau-PET patterns.** Associations between group-average subcortical 18-F-Pi-2620-PET data and seed-based functional connectivity of tau epicenters (i.e., regions with highest group-average tau) in progressive supranuclear palsy—Richardson syndrome (PSP-RS, **A**) and cortico-basal syndrome (CBS, **B**), illustrating that regions with high connectivity to the tau epicenter show high tau-PET. The same association was plotted for tau coldspots (i.e., regions with lowest tau-PET) for PSP-RS (**C**) and CBS (**D**), illustrating that regions closely connected to the tau coldspots show also low tau-PET. Standardized beta- and p-values were derived from linear regression controlling for Euclidean distance between ROIs. The analysis was repeated for all regions of interest (ROIs), and the respective seed ROIs tau-PET uptake was plotted against the regression-derived beta-value, showing that seed regions with higher tau-PET show a positive association between seed-based connectivity and tau-PET in connected regions, whereas regions with lower tau-PET show a negative association between seed-based connectivity and tau-PET in connected regions in PSP (**E**) and CBS (**F**). These findings indicate that seed ROIs are preferentially connected to other regions with similar tau-PET levels. All analyses were repeated including using the combined set of 200 cortical and 32 subcortical ROIs, showing a fully consistent result pattern across the entire brain (**G**–**L**). Two-sided p-values have been determined via linear regression. Linear model fits are indicated together with 95% confidence intervals. Source data are provided as a Source Data file.

intercept. However, for cortical regions, we found that the high Aβ group showed overall higher $^{18}$F-PI-2620 PET binding across cortical Q1 to Q4 ROIs (main effect of Aβ: b/SE = 0.078/0.006, $p = 0.031$, Fig. 6G), yet with the same gradient of tau deposition from tau epicenters throughout connected regions, controlling for age, sex, study site, Euclidean distance to the tau epicenter, and random intercept. This result pattern supports the view that subcortical to cortical tau propagation may be enhanced in the presence of subtle subthreshold Aβ accumulation in CBS.

**Functionally connected brain regions show correlated postmortem tau levels in PSP.** Lastly, we aimed to replicate the association between covariance in $^{18}$F-PI-2620 PET binding and functional connectivity using gold-standard postmortem tau assessments since the in vivo $^{18}$F-PI-2620 signal could still be driven by a parallel phenomenon not directly reflecting tau. The histo-pathological data of our study were derived from regional and cell-type-specific semiquantitative AT8 (Munich sample) or PHF-1 (UPENN sample) stained tau assessments (i.e., for neuronal, astrocyte, and oligodendrocyte tau) in two independent samples with confirmed PSP 4 R tau pathology:[8] Using the neuropatho-logical probe extraction protocols, we reconstructed spatially match-ing bilateral ROIs from established cortical and subcortical anatomical atlases[33, 35], as shown in Fig. 7A for the Munich-European consortium/collection sample ($n = 97$, 16 ROIs) and Fig. 7B for the UPENN sample ($n = 96$, 9 ROIs). For these ROIs, we computed covariance in postmortem stained neuronal tau levels, defined as the partial Spearman correlation between inter-regional neuronal tau, accounting for age at death and gender (Fig. 7C, E). Using the same ROIs shown in Fig. 7A, B, we determined inter-regional functional connectivity in the sample of 69 cognitively normal, amyloid-PET, and tau-PET-negative ADNI subjects (Fig. 7D, F). As for the $^{18}$F-PI-2620 PET analyses, we tested the association between inter-regional functional connectivity and covariance in postmortem tau, focusing particularly on neuronal tau which we hypothesized to be most strongly driven by connectivity-mediated tau spreading. As for $^{18}$F-PI-2620 PET, these analyses were controlled for inter-regional Euclidean distance, to ensure that associations between connectivity and tau covariance were independent of spatial proximity between ROIs. In line with the $^{18}$F-PI-2620 PET data, we found the expected association between functional connectivity and covariance in neuronal tau levels in the Munich-European consortium/collection (β = 0.503, $p < 0.001$, Fig. 7G) and UPENN sample (β = 0.790, $p < 0.001$, Fig. 7H). Again, these associations were confirmed by exact tests ($p < 0.001$, beeswarm plots in Fig. 6G, H), i.e., by comparing the actual β-value with a null-distribution of β-values obtained via repeating the analysis 1000 times using shuffled connectomes with preserved weight- and degree distribution. For direct comparison of the postmortem and tau-PET analyses, we also determined the covariance in tau-PET in PSP-RS patients for the postmortem atlas

regions shown in Fig. 7A, B. In line with the postmortem data, we found a strong association between functional connectivity and covariance in tau-PET in PSP-RS, controlling for Euclidean distance (Munich parcellation [Fig. 7A], β = 0.66, $p < 0.001$; UPENN par-cellation [Fig. 7B], β = 0.52, $p < 0.001$). In the postmortem data, we further tested whether the association between functional con-nectivity and covariance in tau was strongest for neurons. To this end, we recomputed the above-described analyses with measures of neuronal, astroglial, and oligodendroglial tau in 1000 bootstrapped samples that were randomly drawn from the overall samples. Plotting the resulting β-value distributions revealed that the asso-ciations between functional connectivity was indeed highest for neuronal tau, followed by oligodendrocyte and lastly astrocyte tau consistently across the Munich-European consortium/collection (Fig. 7I) and UPENN sample (Fig. 7J). Together, these postmortem findings in two large independent samples replicate the in vivo $^{18}$F-PI-2620 PET findings, showing an association between functional connectivity and covariance in tau pathology, which is strongest for neuronal tau.

## Discussion
Our major aim was to investigate whether functional connectivity is associated with the deposition patterns of tau pathology in 4 R tauopathy patients. To this end, we combined template-based resting-state fMRI connectomics with (i) in vivo tau-PET in 46 PSP-RS and CBS patients and (ii) regional postmortem tau assessments in two independent samples with histopathologically confirmed 4 R tau PSP pathology ($n = 97/n = 96$). Using the next generation tracer $^{18}$F-PI-2620 for imaging tau pathology[27], we found elevated PET binding in clinically diagnosed CBS and PSP-RS patients, particularly in subcortical predilection sites of 4 R tau pathology. Binding of PI-2620 to 4 R tau pathology was further supported by competitive in vitro self-blocking assessments and autoradiography in PSP patient samples. Across CBS and PSP-RS, we show that functionally interconnected subcortical and cortical regions show correlated tau-PET levels. Moreover, we report that patient-level $^{18}$F-PI-2620 PET patterns could be predicted by the seed-based connectivity patterns of subcortical tau epicenters, suggesting that gradual tau aggregation expands from subcortical tau starting sites throughout connected regions. In CBS, which is typically characterized by more widespread cortical tau[23], we show further that subthreshold Aβ-levels are associated with elevated cortical $^{18}$F-PI-2620 PET in regions that are closely connected to the subcortical tau epicenters. By translating the tau-PET vs. connectivity analysis approach to regionally sampled postmortem data from two independent PSP patient samples, we replicated the association between in-vivo-derived inter-regional connectivity and inter-regional covariance in AT8/PHF-1-stained tau pathology. Here, we could show further that the association between connectivity and covariance in postmortem-assessed tau pathology was strongest for neuronal tau compared to glial tau

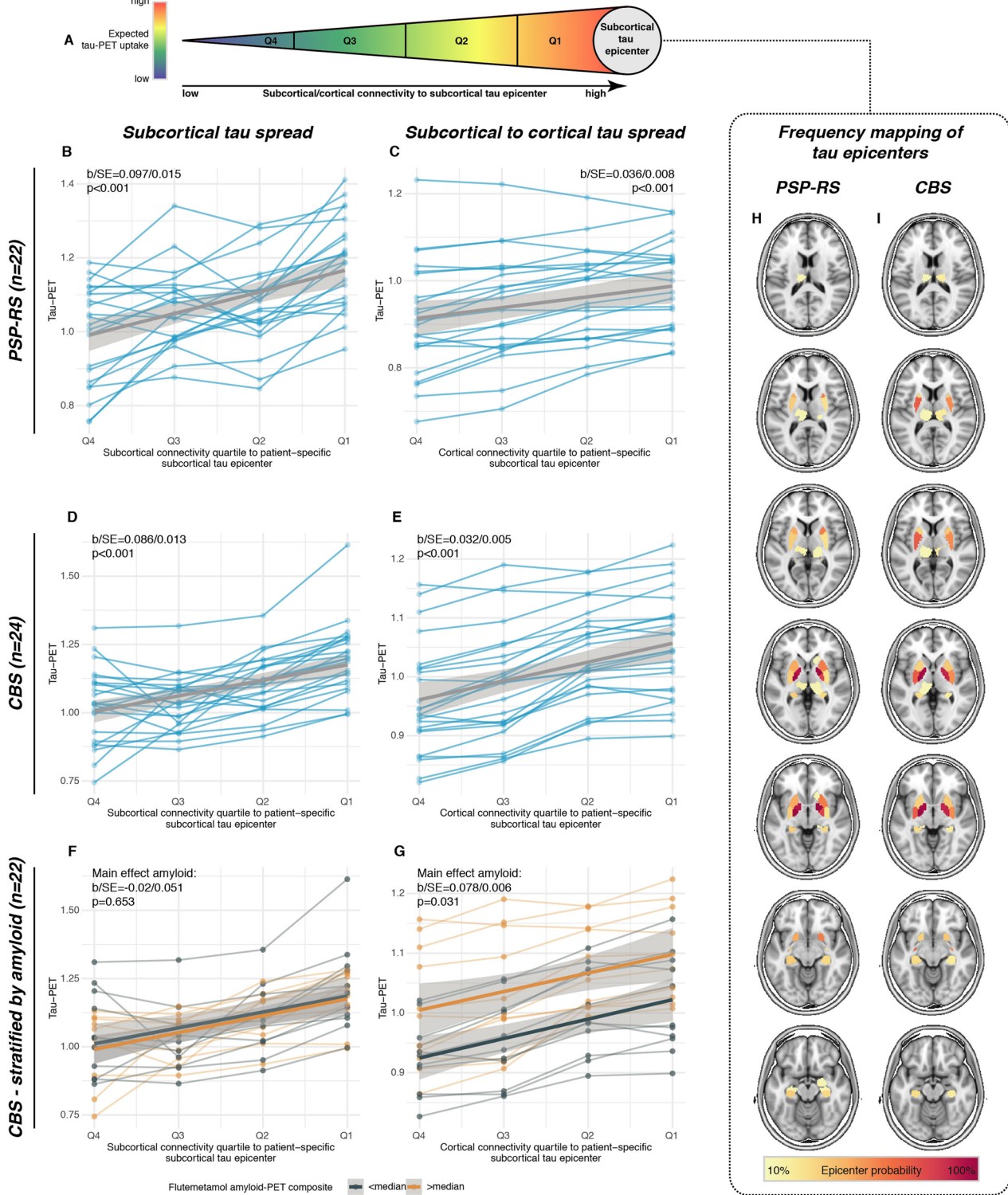

Functional connectivity associated with subject-level tau-PET uptake

levels, supporting the view that connectivity is particularly associated with trans-neuronal tau propagation. Together, our findings provide comprehensive translational evidence for a key role of connectivity in the propagation of 4 R tau pathology[36, 37].

When assessing disease-associated tau-PET patterns, we found elevated [18]F-PI-2620 PET binding in CBS/PSP-RS patients particularly in the basal ganglia, i.e., typical sites of 4 R tau

aggregation[7, 8], congruent with a previous [18]F-PI-2620 PET study in a partly overlapping sample[27]. Further, the epicenters of [18]F-PI-2620 PET spatially matched the earliest signs of tau pathology in the pallido-nigrolusyian axis as detected in postmortem analyses of 4 R tauopathy patients with various clinical phenotypes[8]. CBS patients, which are characterized by a clinically mixed subcortical/cortical phenotype, showed more widespread cortical

**Fig. 6 Subject-level epicenter connectivity vs. tau-PET patterns.** Using subject-level tau-PET data, we determined for each progressive supranuclear palsy —Richardson syndrome (PSP-RS) and cortico-basal syndrome (CBS) patient the subcortical tau epicenter (**A**), i.e., defined as 20% of ROIs with highest tau-PET SUVRs. The remaining regions of interest (ROIs) were grouped for each subject into quartiles, depending on connectivity strength to the subject-specific tau epicenter. Highest tau-PET was expected for regions most closely connected to the tau epicenter (i.e., quartile 1 = Q1) whereas lowest tau-PET was expected for ROIs only weakly connected to the tau epicenter. Subject-specific tau-PET data for subcortical Q1–Q4 ROIs (**B**, **D**) as well cortical Q1–Q4 ROIs (**C**, **E**) is shown, illustrating that tau-PET was highest in subcortical and cortical regions that are most closely connected to the subcortical tau epicenter (i.e., Q1), with gradual decreases across less strongly connected regions. For a subset of CBS patients ($n = 22$), we further stratified these analyses by above vs. below median global amyloid-PET SUVRs (i.e., subthreshold amyloid, as all subjects were amyloid negative on visual read). Amyloid-stratified analyses illustrate that above median amyloid levels were not associated with elevated tau spread from subcortical epicenters to subcortical Q1–Q4 ROIs (**F**), but with increased tau spread from subcortical epicenters to cortical Q1–Q4 ROIs (**G**). All statistical indices (i.e., b-values, standard errors and p-values) were derived from linear mixed models, controlling for age, sex, study center, mean Euclidean distance of Q1–Q4 ROIs to the tau epicenter and random intercept. A probability mapping of subcortical epicenter locations is illustrated in panels **H** for PSP-RS and **I** for CBS patients. Two-sided p-values have been determined via linear mixed effects models. Linear model fits are indicated together with 95% confidence intervals. Source data are provided as a Source Data file.

[18]F-PI-2620 PET elevations than PSP-RS, in line with post-mortem observations[23]. Together, the [18]F-PI-2620 PET patterns observed in PSP-RS and CBS are congruent with histopathologically observed 4 R tau patterns in CBD/PSP[7, 8].

Nevertheless, we note as a limitation that in vivo and postmortem data were not assessed in the same patients in the current study. Thus, although we and others provide autoradiographic evidence for [18]F-PI-2620 binding to 4 R tau[27, 38], the observed signal could still derive from a source closely paralleling 4 R tau. Regarding our main finding, we observed that functionally connected brain regions exhibit correlated [18]F-PI-2620 PET levels in CBS and PSP-RS. This result echoes previous evidence of covarying cross-sectional tau-PET levels and longitudinal tau-PET accumulation rates among functionally interconnected regions in AD, suggesting a consistent association between tau deposition patterns and brain connectivity across different tauopathies[39–41]. This interpretation is also supported by preclinical work, showing that pathological tau species from 4-repeat (i.e., CBD/PSP) and mixed 3/4-repeat tauopathies (i.e., AD) consistently trigger transcellular tau spread across connected brain regions while recapitulating disease-specific neuropathological hallmark features (e.g., neuronal tau in AD vs. neuronal and glial tau in CBD/PSP)[15, 17, 42]. Recent studies have emphasized, however, that CBD and PSP tau show distinguishable molecular characteristics[43–45], hence it will be important in future studies to assess whether molecular differences in CBD and PSP tau modulate the spreading potential of tau pathology. Importantly, we found an association between connectivity and covariance in [18]F-PI-2620 PET binding not only for subcortical 4 R tau predilection sites but also for cortico-cortical and subcortico-cortical connections. This indicates that associations between connectivity and tau deposition patterns are not spatially restricted to particularly vulnerable brain regions, in line with previous evidence in AD patients[34, 39, 40] and preclinical tauopathy models[15, 17, 42].

In a critical validation step, we were able to translate these in vivo tau-PET vs. connectivity analyses to postmortem tau assessments in two pre-existing datasets of patients with histopathologically confirmed PSP[8]. In these unique samples with gold-standard regional tau assessments, we found a highly consistent result pattern with strong associations between in-vivo-assessed inter-regional functional connectivity and inter-regional correlations of AT8 or PHF-1 stained tau levels. Importantly, these associations were replicated across both independent postmortem samples with regionally different postmortem sampling (see Fig. 7A, B), supporting the robustness of our findings. Even more important, cell-type-specific tau assessments confirmed the hypothesis that the association between connectivity and covariance in tau was strongest for neuronal tau pathology when compared to astroglial and oligodendrocyte-specific tau, thus supporting neuronal connectivity as the main driver of

trans-synaptic tau propagation[22]. Consistent across both postmortem samples, the second strongest association between connectivity and tau covariance was observed for oligodendrocytes, i.e., the main constituents of myelin sheaths along axonal tracts. Previous work in mice has shown that 4 R oligodendrocyte tau propagates along white matter tracts, even in the absence of neuronal tau pathology, suggesting oligodendrocyte tau propagation along anatomically interconnected pathways[46]. A recent study evaluating cell-specific sequences showed more overlap of the patterns of neuronal and oligodendrocytic tau deposition, and albeit early steps of astrocytic tau deposition deviates from the neuronal, it converges in later steps[8]. Astrocytic tau accumulation may be seen without local neuronal tau pathology in regions with high connectivity to regions with existing neuronal tau pathology[47], which together with a preclinical study that observed astrocytic tau pathology only in the presence of neuronal tau[46], are in line with the observation that astrocytes phagocytose neuronally released tau[15], either from local neurons or projecting from other regions[48]. Altogether, these may explain why connectivity and astrocytic tau were associated, albeit to a weaker extent compared to neuronal and oligodendrocytic tau. Thus, these preclinical findings converge with our result pattern of strongest associations between connectivity and neuronal tau, followed by oligodendrocyte and astrocyte tau. In summary, our findings provide compelling postmortem replication of an association between connectivity and 4 R tau spreading.

Importantly, all associations between connectivity and covariance in postmortem tau or in vivo [18]F-PI-2620 PET were detected while statistically controlling for inter-regional Euclidian distance, highlighting that the association between tau and connectivity is not driven by mere spatial proximity. This result pattern supports the view that tau spread is driven by connectivity and not proximity[21, 49, 50], in line with preclinical studies, showing that cerebral tau injections trigger tau spread to connected rather than spatially adjacent regions[15, 42].

In the [18]F-PI-2620 PET sample, we further show that brain regions with highest tau (i.e., tau epicenters) are closely functionally connected to other high tau regions, whereas low tau regions are most closely connected to other low tau regions. These findings are congruent with postmortem tau staging schemes, showing that 4 R tau pathology is initially confined to circumscribed epicenters from where it progresses gradually throughout the brain[8]. Further supporting this, we found that connectivity patterns of subject-level tau-PET epicenters predicted a gradient of tau deposition, with highest tau in subcortical/cortical regions that were most closely connected to the tau epicenter, while tau levels were gradually lower in farther connectivity-based distance to the tau epicenter. In line with findings in other tauopathies such as AD[34, 39, 41, 51], these results suggest tau spreading patterns in 4 R tauopathies are determined

**Functional connectivity associated with covariance in post-mortem tau assessments**

by the connectivity pattern of the tau epicenter, thereby strongly supporting the concept of connectivity-based tau spreading. Importantly tau spreading concepts applied to PET imaging could facilitate patient-tailored precision medicine since they may facilitate prediction of the future accumulation pattern of tau pathology and the progression/development of symptoms on a patient-level[34]. By implication, identifying sites that are

vulnerable to tau spreading could also serve as a imaging-guided primary read-out in tau targeting trials of 4 R tauopathies.

In an exploratory subanalysis in CBS patients with available global amyloid-PET data, we found further that subthreshold Aβ-levels were associated with higher cortical tau-PET-levels following the same connectivity gradient of subcortical tau epicenters (Fig. 5G). A potential explanation is that subtly elevated

**Fig. 7 Connectivity vs. Post-mortem tau deposition patterns.** Association between functional connectivity and regional postmortem tau assessments (i.e., AT8 staining) in two independent samples with histopathologically confirmed progressive supranuclear palsy (PSP = . For each sample, covariance in neuronal tau levels was assessed among cortical and subcortical ROIs (**A**, **B**) using the methods illustrated in Fig. 3A, yielding a covariance in AT8-stained tau matrix of partial correlations accounting for age at death and sex. for the Munich (**C**) and UPENN sample (**E**). Using the same brain atlases (**A**, **B**), functional connectivity was determined based on resting-state fMRI data in the sample of $n = 69$ healthy controls from ADNI (**D**, **F**). Scatterplots illustrate the association between functional connectivity and covariance in postmortem stained neuronal tau pathology for the Munich (**G**, AT8 tau staining) and UPENN (**H**, PHF-1 tau staining) sample. Standardized beta- and p-values were derived from linear regression controlling for Euclidean distance between ROIs. Robustness of the association in panels **D**, **E** was again tested by contrasting the beta-value derived from the association between the actual functional connectivity matrix with covariance in tau against beta-values derived from the same analyses repeated a 1000 times using scrambled functional connectivity matrices with preserved degree- and weight distribution (see beeswarm plots in panels **G**, **H**). The same analysis was repeated for cell-specific tau levels (i.e., astrocyte tau, oligodendrocyte tau, neuronal tau) using 1000 bootstrapping iterations, i.e., the association between functional connectivity and covariance in cell-specific tau was repeated on 1000 randomly drawn samples. The resulting beta-value distributions are shown for the Munich (**I**) and UPENN (**J**) sample, illustrating that the association between functional connectivity and covariance in tau is strongest for neuronal tau levels. Two-sided p-values have been determined via linear models for scatterplots. Linear model fits in scatterplots are indicated together with 95% confidence intervals. Boxplots are displayed as median (center line) ± interquartile range (box boundaries) with whiskers including observations falling within the 1.5 interquartile range. Two-sided p-values in boxplots have been obtained using ANOVAs. Source data are provided as a Source Data file.

Aβ levels (i.e., all CBS patients were by definition Aβ-negative), which have been previously shown to accelerate AD-typical tau spreading[52], also accelerate subcortical to cortical tau spreading in CBS patients. As a limitation, we note that different binding characteristics of [18]F-PI-2620 to 3/4 R and 4 R tau isoforms could also explain our observation since the Aβ status impacts the predominant tau isoform and tau isoform shifts[53]. Yet, this finding is preliminary and requires further systematical investigation by larger dedicated studies and longitudinal analyses. In addition, a potential association between cortical Aβ-levels and elevated cortical tau-PET should be explored in PSP cases, especially in those with cortical phenotypes (e.g., language variant PSP) once sufficient data become available.

Several limitations should be considered when interpreting our results. First, the [18]F-PI-2620 PET cohort includes PSP-RS and CBS patients with clinically diagnosed 4 R tauopathies, since biomarker-based 4 R tauopathy diagnosis is not yet clinically established[12, 28]. Further, PI-2620 controls were significantly younger than PSP-RS patients, hence age-related increase in tau-PET signal may potentially bias group comparisons between controls and PSP-RS groups, despite age adjustment of voxel-wise analyses. Importantly, however, [18]F-PI-2620 PET patterns in PSP-RS and CBS groups matched the tau patterns that are typically observed in histopathological assessments[7, 8], and subcortical [18]F-PI-2620 PET was significantly higher relative to controls, thus supporting the view that the [18]F-PI-2620 PET signal in these 4 R tau predilection sites reflects no off-target signal but an increase in tau pathology[27]. This is supported by our in vitro binding assay data, showing a stronger affinity of PI-2620 to 4 R tau fibrils than Flortaucipir (Fig. 2A), as well as the postmortem autoradiography assessments in PSP patients, showing that stronger PI-2620 signal is observed in brain regions with higher AT8-stained tau pathology (Fig. 2C, D). Further, a recent study refined PI-2620 binding sites to tau from cryo-EM metadynamics simulations in order to provide atomic resolution of the binding modes and thermodynamic properties[54]. Here, several binding sites for PI-2620 were observed on 4 R Tau CBD fibrils which support sufficient PI-2620 in vivo binding to 4 R tau. Nevertheless, the affinity of [18]F-PI-2620 PET to 4 R tau needs to be characterized further and our findings warrant replication with other candidate tracers for 4 R tau (e.g., [18]F-PM-PBB3)[55]. In addition, [18]F-PI2620 shows off-target binding to neuromelanin[38], hence tau deposition in neuromelanin harboring brainstem regions, which typically show early tau accumulation in 4 R tauopathies[8], can currently not be assessed with [18]F-PI-2620 PET. To further guard against the impact of confounding 3/4 R pathology, all patients with CBS were rated Aβ negative as

determined via cerebrospinal fluid or amyloid-PET assessments, suggesting that elevated [18]F-PI-2620 PET is not primarily explained by AD-typical 3 R/4 R tau[56, 57]. Second, the current study focused on functional connectivity, i.e., shared inter-regional BOLD activity, which is to a large degree but not entirely matched by structural connectivity as assessed via diffusion MRI[58]. This structural/functional connectivity mismatch is partly determined by technical limitations of MRI-based tractography to detect crossing-fibers or short-range cortico-cortical connections[59]. On the other hand, the slow temporal resolution of fMRI may introduce connectivity between regions without direct but rather indirect multi-synaptic connections[60]. Thus, our results on tau covariance vs. connectivity likely reflect a mixture of direct and indirect connections. In addition, the fMRI BOLD signal stems from multiple cellular sources including astrocytes, neurons, and vasculature[61], and while fMRI-assessed connectivity is associated with electrophysiological brain activity[62], fMRI connectivity does potentially not fully match neuronal connectivity. However, the advantage of functional MRI in the current study is the possibility to map a proxy of brain connectivity between spatially adjacent subcortical nuclei, where connections are not accessible with diffusion-based tractography. In addition, the current study used connectivity templates derived from healthy individuals. Thus, it remains to be determined by future studies whether subject-specific connectivity differences (i.e., connectome fingerprinting) contribute to heterogeneity in tau spreading patterns. Here, it will also be important to assess whether disease-related connectivity changes, such as tau-related disruptions in functional connectivity further modulate downstream tau spread that deviates from tau spreading patterns modeled with a healthy connectome template. Third, the current study is cross-sectional in nature and thus does not assess the association between connectivity and the spatio-temporal progression of tau pathology. In order to test the predictive value of connectivity for regional changes in tau-PET, longitudinal tau-PET studies are necessary, which will be conducted as soon as large enough data are available.

In conclusion, the current study demonstrates a close link between 4 R tau deposition patterns and connectivity, thereby supporting the concept of trans-neuronal tau spreading in 4 R tauopathies[13, 22, 37]. A clear strength of the current study is the translational study design with independent validation of the association between inter-regional functional connectivity and covariance in tau pathology across in vivo [18]F-PI-2620 PET and cell-specific postmortem histopathological tau assessments in multiple independent samples. The current results may be used as a starting point for future studies with detailed phenotyping of

cognitive and motor function to map tau spreading patterns and downstream neurodegeneration to cognitive and motor phenotypes in patients with 4 R tauopathies. Since tau pathology is closely assumed to be a key driver of disease progression[5, 7, 8], our results suggest that interventions that target tau spreading are potentially key therapeutic strategies.

## Methods

**Tau-PET sample**. We included 61 subjects recruited at four sites (Munich & Leipzig, Germany; Melbourne, Australia; New Haven, United States), including 15 cognitively normal individuals (i.e., without evidence of cognitive decline, any motor symptoms or cerebral tau pathology), 24 patients with clinical diagnosis of possible or probable cortico-basal syndrome (CBS) and 22 patients with clinical diagnosis of PSP-RS. CBS diagnosis was made according to the revised Armstrong Criteria of probable CBS[12] or the Movement Disorders Society criteria of possible PSP with predominant CBS[28]. PSP-RS was diagnosed following state-of-the-art diagnostic criteria[28]. All patient data derive from a PSP cohort recruited in Munich and Leipzig[27] and a CBS cohort recruited in Munich[31]. Inclusion criteria for the current study were age above 45 years, stable pharmacotherapy for at least 1 week before the tau-PET examination, negative family history for Parkinson's and Alzheimer's disease and availability of 3D T1-weighted structural MRI. Exclusion criteria were severe neurological or psychiatric disorders other than PSP and CBS or positive Aβ status, as determined via expert visual read of 18-F-Flutemetamol or 18-F-florbetaben amyloid-PET or by cerebrospinal fluid analyses of Aβ levels using locally established cut-offs (i.e., $A\beta_{42/40}$-ratio < 5.5% or $A\beta_{1-42}$ < 375 pg/ml). The 18-F-PI2620 PET imaging protocol was approved by local ethics committee of the LMU Munich. Written informed consent was obtained from all participants in accordance with the Declaration of Helsinki. The full study protocol including all samples and all PET data analyses were approved by the local ethics committee (LMU Munich, application numbers 17-569 and 19-022) and the German radiation protection (BfS-application: Z 5 − 22464/2017-047-K-G) authorities. The study was carried out according to the principles of the Helsinki Declaration, patients received no compensation for study participation. All work complied with ethical regulations for work with human participants.

**Neuroimaging acquisition**. All structural MRI data were collected on 3 T scanners using 3D MPRAGE and MP2RAGE sequences. Radiosynthesis of 18-F-PI-2620 was achieved by nucleophilic substitution on a butyloxycarbonyl-protected nitro precursor using an automated synthesis module (Synthera, IBA Louvain-la-neuve, Belgium). The protecting group was cleaved under radiolabeling conditions. The product was purified by semipreparative high performance liquid chromatography, resulting in radiochemical purity of >97%. Non-decay corrected yields were about 30% with a molar activity of $3 \cdot 10^6$ GBq/mmol at the end of synthesis. 18-F-PI-2620 PET imaging in combination with computed tomography (CT) or magnetic resonance (MR) was performed in a full dynamic setting (minimum scan duration: 0–60 min post-injection) using pre-established standard PET scanning parameters at each site: In Munich, Germany, dynamic tau-PET was acquired on a Siemens Biograph True point 64 PET/CT (Siemens, Erlangen, Germany) or a Siemens mCT scanner (Siemens, Erlangen, Germany) in 3D list-mode over 60 min and reconstructed into a $336 \times 336 \times 109$ matrix (voxel size: $1.02 \times 1.02 \times 2.03$ mm³) using the built-in ordered subset expectation maximization (OSEM) algorithm with 4 iterations, 21 subsets, and a 5 mm Gaussian filter. A low dose CT was used for attenuation correction. In Leipzig, Germany, dynamic tau-PET was acquired on a hybrid PET/MR system (Biograph mMR, Siemens Healthineers, Erlangen, Germany) in 3D list-mode over 60 min and reconstructed into a $256 \times 256$ matrix (voxel size: $1.00 \times 1.00 \times 2.03$ mm³) using the built-in ordered subset expectation maximization algorithm with 8 iterations, 21 subsets, and a 3 mm Gaussian filter. For attenuation correction, the vendor-provided HiRes method was employed, combining the individual Dixon attenuation correction approach with a bone attenuation template[63]. For tau-PET data of control subjects imaged in New Haven, US, dynamic PET was acquired on a Siemens ECAT EXACT HR + camera from 0 to 90 and 120 to 180 min. Images were reconstructed in a $128 \times 128$ matrix (zoom = 2, pixel size of $2.574 \times 2.574$ mm) with an iterative reconstruction algorithm (OSEM 4 iterations, 16 subsets) and a post-hoc 5 mm Gaussian filter. Standard corrections for random, scatter, system dead time and attenuation provided by the camera manufacturer were performed. 18-F-PI-2620 PET assessments of control subjects in Melbourne was performed on a Philips Gemini TF 64 PET/CT (Philips, Eindhoven, The Netherlands). PET images were acquired dynamically from 0 to 60 min and 80 to 120 min post-injection. Images were reconstructed using LOR-RAMLA and CT attenuation correction was performed. Images were binned into a $128 \times 128 \times 89$ matrix (voxel size: $2.00 \times 2.00 \times 2.00$ mm³).

The injected dose was 168–334 MBq, applied as a bolus injection. Site-specific attenuation correction ensured multi-site harmonization of data acquisition[63]. Further, data from Hofmann brain phantoms were used to obtain scanner-specific filter functions which were consequently used to generate images with a similar spatial resolution for voxel-wise analyses (full-width-at-half-maximum = $9 \times 9 \times 10$ mm; determined by the scanner in New Haven), following the Alzheimer's Disease Neuroimaging Initiative image harmonization procedure[64]. Resulting smoothing factors were $3.5 \times 3.5 \times 7.0$ mm for Munich,

6.0 × 6.0 × 6.0 mm for Leipzig, and 4.0 × 4.0 × 4.0mm for Melbourne. All dynamic datasets were visually checked for artifacts and motion-corrected using rigid-registration. Mean SUV images were obtained for 20–40 min time frames to obtain SUVR images, which show comparable performance in signal sensitivity and were less subject to artifacts when compared to 0–60 min or 0–40 min DVR images[65].

**Structural MRI and tau-PET preprocessing**. All structural MRI and PET data were processed using the Advanced Normalization Tools (ANTs) toolbox (http://stnava.github.io/ANTs/). In an initial step, 18-F-PI-2620 PET images were rigidly co-registered to native-space T1-weighted MRI images. For T1-weighted structural MRI data we performed bias field correction, brain extraction, and segmentation into gray-matter, white matter and cerebrospinal fluid tissue maps using the ANTs cortical thickness pipeline. Brain extracted T1-weighted images were nonlinearly normalized to MNI space (2 mm isotropic voxels) using ANTs high-dimensional warping algorithm[66]. By combining the rigid 18-F-PI-2620 PET to T1-native space and nonlinear T1 to MNI space spatial normalization parameters, all brain-atlas-derived ROI data was transformed from MNI space back to PET native space, including the Tian 32 ROI subcortical brain atlas[33] (Fig. 1H), the Schaefer 200 ROI cortical brain atlas[32] (Fig. 1G), as well as the inferior cerebellar reference ROI for intensity normalization of tau-PET[67]. All brain-atlas data and the inferior cerebellar reference ROI were further masked with binary subject-specific gray-matter maps in order to restrict later extraction of ROI-mean values to gray-matter regions.

Tau-PET images were intensity normalized to the mean tracer uptake of the inferior cerebellar gray, to determine standardized uptake value ratio (SUVR). Mean tau-PET SUVR values were extracted for each subject for the 32 subcortical and 200 cortical ROIs from unsmoothed native-space PET data. For voxel-wise analyses, subject-specific tau-PET SUVR images were warped to MNI space by combining the linear PET to T1 and nonlinear T1 to MNI transformation parameters, followed by spatial smoothing with site-specific smoothing kernels. Usage of an alternative reference ROI (i.e., eroded white matter) yielded congruent analyses with those presented in the manuscript. To determine whether ROI-based estimation of tau-PET ROIs was biased by gray-matter atrophy, we further obtained partial-volume effect (PVE) corrected tau-PET SUVRs, using the Gaussian transfer method[68].

**Postmortem sample**. To replicate tau-PET vs. connectivity associations using postmortem assessments of tau pathology, we included histopathological tau data from two independent samples, including $n = 97$ PSP subjects recruited, sampled, and examined at several different sites across Europe with centralized final analysis at the Department of Neuropathology, LMU, in Munich and $n = 96$ PSP subjects from the University of Pennsylvania. An in-depth description of data selection and data acquisition has been published previously[8]. Cases were selected based on presence of neurofibrillary tangles in the subthalamic nucleus, substantia nigra, and pallidum, as well as based on the presence of tufted astrocytes in the striatum and frontal cortex[3, 69]. All donors or their next of a kin had given written informed consent according to the Declaration of Helsinki for the use of brain tissue and medical records for research purposes. Usage of the material was in accordance with the directives of local ethics commissions regarding the use of archive material for research purposes. The postmortem tau histopathological examinations of brain tissue in PSP patients were approved by the ethics committee of the medical faculty of the university of Marburg, Germany.

Extraction of neuropathological samples followed a standardized pre-established procedure[8]. Formalin-fixed and paraffin-embedded tissue blocks from the PSP cases were evaluated using tau immunostaining with the anti-tau AT8 antibody (Ser202/Thr205, 1:200, Invitrogen/Thermofischer, MN1020, Carlsbad, USA) for the Munich sample, and with anti-tau PHF-1 (Ser396/Ser404, 1:2000) for the UPENN sample. For each sample, we included data from postmortem sampled regions of interest, which were judged accessible with functional MRI, resulting in 16 ROIs for the Munich sample and 9 ROIs for the UPENN sample. Neuropathological ROIs were reconstructed in MNI space based on the neuropathological examination protocol using predefined ROIs from established anatomical brain atlases (Fig. 6A, B)[33, 35]. Neuronal tangle pathology was graded for each region in a semiquantitative score (none = 0, mild = 1, moderate = 2, severe = 3).

For direct comparison with the autoradiography signal, tau immunostaining from formalin-fixed and paraffin-embedded tissue blocks from 16 PSP cases and three brain regions (frontal cortex, putamen, and pallidum) was processed and graded as described above. For each patient and brain region, autoradiography with 18-F-PI-2620 was performed on ≥4 sections (Superiority of Formalin-Fixed Paraffin-Embedded Brain Tissue for in vitro Assessment of Progressive Supranuclear Palsy Tau Pathology With [18 F]PI-2620, Willroider et al. 2021). In short, sections were incubated for 45 min (21.6 µCi/ml after dilution to a volume of 50 ml with phosphate buffered saline solution, pH 7.4, specific activity 480 ± 90 GBq/µmol), washed, dried, placed on imaging plates for 12 h and scanned at 25.0 µm resolution. Regions of interest were drawn on each sample with AT8 staining of adjacent section serving for precise anatomical definition with AT8-negative white matter as reference region. Binding ratios were correlated with semiquantitative AT8 assessment using spearman correlation and differences between groups were assessed using ANOVA.

**In vitro competition assay**. K18 fibrils, depicting 4 R tau deposits (~5 μg protein per well) were incubated with $^{18}$F-PI-2620 or $^{18}$F-AV1451 (0.7 kBq per well) and the respective cold compound(s) ranging from 0.61 nM to 1000 nM for 45 min at 37 °C in a 96-well plate. The assay was performed in a total volume of 200 μL PBS w/o Ca + +/Mg + + containing 0.1% BSA and 2% DMSO. Nonspecific signal was determined with samples containing $^{18}$F-labeled tau-tracer in the presence of assay buffer w/o brain substrate and competitor. The assay controls without brain homogenate were incubated with the tracer in parallel. After 45 min incubation, the samples were filtered under vacuum on a GF/B filter plate (PerkinElmer 6005177). The GF/B filters were equilibrated with PBS (w/o Ca + +/Mg + +, 0.1% BSA, 2% DMSO) at least 1 h before filtration. Afterwards, the filters were washed two times with 200 μL PBS (w/o Ca + +/ Mg + +, 0.1 % BSA, 2% DMSO). The top and bottom side of the filter plates were sealed and a phospho-imaging plate was placed on top of the filter plate and exposed overnight. The imaging plate was scanned using the BASReader 5000 (Fuji) and quantified with the AIDA software. Specific binding was calculated by subtracting the nonspecific signal from the measured sample signals. The unblocked $^{18}$F-labeled tracer signal was defined as total binding. $IC_{50}$ values were calculated using Prism V8.

**Assessment of covariance in tau-PET and postmortem tau**. We assessed the inter-regional covariance in 18-F-PI-2620 PET SUVRs (see Fig. 2A for an analysis flow-chart) for 22 PSP-RS and 24 CBS patients. The analysis pipeline was adopted from previous studies using this approach to determine FDG-PET covariance (i.e., metabolic covariance), gray-matter covariance (i.e., structural connectivity), or 18-F-AV1451 PET covariance across brain regions[39, 40, 70, 71]. First, we computed the mean tau-PET uptake within each of the 200 cortical (Fig. 1G) and 32 subcortical (Fig. 1H) ROIs for each subject (Fig. 2, A(i)). Next, we vectorized mean ROI SUVR values to subject-specific 232-element vectors (Fig. 2, A(ii)). Using these 232-element 18-F-PI-2620 PET SUVR vectors, we then assessed across subjects the pairwise ROI-to-ROI partial correlation of 18-F-PI-2620 PET uptake (Fig. 2, A(iii)), adjusting for subject-specific age, gender, and PET imaging and MRI protocol as potential confounds. This analysis resulted in a 232 × 232 sized 18-F-PI-2620 PET covariance matrix each for each group (i.e., PSP, CBS). Within this 18-F-PI-2620 PET covariance matrix, autocorrelations were set to zero and all correlations were Fisher-z transformed.

An equivalent approach was used for postmortem tau data. For each sample, we determined the inter-regional partial Spearman correlation of postmortem AT8-stained tau pathology across the 16 ROIs of the Munich-European consortium/collection sample (Fig. 6A) or 9 ROIs of the UPENN sample (Fig. 6B), adjusting for age at death and gender, yielding a covariance in postmortem tau matrix (Fig. 6C, E). Note that we specifically used partial Spearman correlation since postmortem ratings were done on an ordinal scale. Again, autocorrelations within this matrix were set to zero and all remaining correlations were Fisher-z transformed.

**Functional connectivity assessment**. For assessing functional connectivity, we used resting-state fMRI data from 69 cognitively normal controls of the ADNI cohort. Ethics approval was obtained by ADNI investigators at participating ADNI sites, all study participants provided written informed consent. These subjects were selected based on absence of objective or subjective signs of cognitive impairment and had no evidence of clinically relevant cerebral amyloid or tau pathology, as indicated by negative $^{18}$F-florbetapir amyloid-PET (i.e., global SUVR < 1.11)[72] and negative $^{18}$F-flortaucipir tau-PET scans (i.e., global SUVR < 1.3)[29].

MRI scans were obtained on Siemens scanners using unified scanning protocols. T1-weighted structural MRI was recorded using a MPRAGE sequence with 1 mm isotropic voxel-space and a TR = 2300 ms. For functional MRI, for each subject a total of 200 resting-state fMRI volumes were recorded using a 3D echo-planar imaging (EPI) sequence in 3.4 mm isotropic voxel resolution with a TR/TE/flip angle = 3000/30/90°.

All images were inspected for artifacts prior to preprocessing. Using ANTs, T1-weighted structural MRI images were bias corrected, brain extracted, segmented and nonlinearly spatially normalized to MNI space[66]. Functional EPI images were slice-time and motion-corrected (i.e., realignment to the first volume) and co-registered to the native T1 images. Using rigid-transformation parameters, T1-derived gray-matter, white matter, and cerebrospinal fluid segments were transformed to EPI space. To denoise the EPI images, we regressed out nuisance covariates (i.e., average white matter and cerebrospinal fluid signal and motion parameters estimated during motion correction), removed the linear trend and applied band-pass filtering with a 0.01–0.08 Hz frequency band in EPI native space. To further minimize the impact of motion which may compromise FC assessment[73], we performed motion scrubbing, where we censored volumes that showed a frame-wise displacement of >1 mm, as well as one prior and two subsequent volumes. In line with our previous work[74], only subjects for whom less than 30% of volumes had to be censored were included in the current study. Note that we did not spatially smooth the functional images during preprocessing to avoid signal spill-over between adjacent brain regions that may artificially enhance functional connectivity between adjacent brain regions during ROI-based connectivity assessment.

To determine functional connectivity, we warped the 232 cortical and subcortical ROIs for tau-PET analyses (Fig. 1G, H) as well as the postmortem ROIs for the Munich-European consortium/collection and UPENN sample (Fig. 6A, B)

to the denoised and preprocessed fMRI images in native EPI space, by combining the linear EPI to T1 and nonlinear T1 to MNI transformation parameters. ROI maps in EPI space were masked with subject-specific gray matter. Fisher-z transformed Pearson-Moment correlations between time-series averaged across voxels within an ROI were determined to assess subject-specific functional connectivity matrices. Functional connectivity data were averaged across all 69 ADNI subjects in order to determine group-average functional connectivity matrices for the in vivo tau-PET analyses as well as for the postmortem analyses.

For the group-average functional connectivity matrices, we further determined 1000 null-models of functional connectivity respectively, by shuffling the functional connectivity matrices while preserving the overall degree- and weight-distribution, using the null_model_und_sign.m function of the brain connectivity toolbox (https://sites.google.com/site/bctnet/).

**Statistics**. Sample demographics were compared between the groups using ANOVAs for continuous measures and Chi-squared tests for categorical measures. Voxel-wise comparisons in 18-F-PI-2620 PET SUVRs were determined on spatially normalized and smoothed tau-PET images using ANCOVAs in SPM12, controlling for age, gender, and study site, applying a voxel-wise alpha-threshold of 0.005 and a cluster-extent threshold of >100 spatially contiguous voxels.

To test the association between functional connectivity and covariance in 18-F-PI-2620 PET, we used linear regression with covariance in 18-F-PI-2620 PET as a dependent variable and ADNI-derived group-average functional connectivity as an independent variable, controlling for inter-regional Euclidean distance (i.e., distance between ROI-specific centers of mass). This analysis was stratified by CBS and PSP-RS and conducted for subcortical ROIs only (i.e., primary analysis) as well as for the whole brain (i.e., secondary analysis). The same analysis was performed for postmortem tau assessments in the Munich-European consortium/collection and UPENN dataset, using regional semiquantitative tau data (i.e., 16 ROIs in the Munich sample vs. 9 ROIs in the UPENN sample) rather than tau-PET. To determine the robustness of the association between connectivity and covariance in 18-F-PI-2620 PET/postmortem tau, the analysis was repeated 1000 times using shuffled connectomes with preserved weight- and degree distribution to obtain a distribution of null-model β-values. We then performed an exact test, i.e., we determined the probability of null-distribution derived β-values surpassing the true β-value. For postmortem data, we further assessed cell-type-specific associations between tau covariance (i.e., neuronal, astroglial and oligodendroglial tau covariance) and functional connectivity. To this end we computed regression-based association between covariance in tau and functional connectivity for each cell type, based on 1000 bootstrapped samples that were randomly drawn from the respective sample (i.e., UPENN or Munich). The resulting β-value distributions reflecting the association between connectivity and cell-type-specific tau covariance were compared between cell-types using paired t-tests.

Using 18-F-PI-2620 PET, we further tested, whether functional connectivity patterns of tau epicenters (i.e., ROIs with highest tau) are predictive of 18-F-PI-2620 PET binding in remaining brain regions. Again, this analysis was determined stratified by group (i.e., PSP-RS vs. CBS) and conducted for subcortical ROIs only (i.e., primary analysis) as well as for the whole brain parcellation (i.e., secondary analysis). Specifically, we determined group-level 18-F-PI-2620 PET binding, and tested whether higher seed-based connectivity of the epicenter ROI was associated with higher 18-F-PI-2620 PET binding in the remaining ROIs, using linear regression controlling for between-ROI Euclidean distance. The same analysis was performed for the ROI with lowest 18-F-PI-2620 PET binding (i.e., coldspot), for which we assumed that higher connectivity is associated with lower 18-F-PI-2620 PET binding in the remaining regions. In an iterative next step, we repeated this analysis across all ROIs and determined the association between the 18-F-PI-2620 PET level in the seed ROI against the regression-derived β-value of the association between seed-based connectivity and 18-F-PI-2620 PET binding in remaining ROIs. As in our previous studies[39, 40], we expected that ROIs with higher tau-PET binding should be connected to other ROIs with a high binding level (i.e., as reflected in a positive β-value), whereas regions with low tau-PET binding should be connected to other ROIs with a low binding level (i.e., as reflected in a negative β-value).

For subject-level analyses, we adopted our pre-established approach[34] and determined subject-level epicenters as 20% of ROIs with highest tau-PET binding (see Fig. 5H, I for tau epicenter probability maps in PSP-RS and CBS). The remaining non-epicenter ROIs were grouped for each subject into 4 quartiles, based on the connectivity strength to the epicenter (i.e., ROIs grouped in Q1 show highest connected to the tau epicenter, vs. ROIs grouped in Q4 show weakest connectivity to tau epicenters). Note that Q1–Q4 ROIs were determined separately for subcortical and cortical regions. Mean 18-F-PI-2620 PET binding was assessed for each subjects' subcortical/cortical Q1–Q4 ROIs. Using linear mixed models, we then tested the association between connectivity strength (i.e., Q1–Q4) and subcortical or cortical 18-F-PI-2620 PET binding in Q1–Q4 ROIs, controlling for age, sex, study site, mean Euclidean distance to the epicenter as well as random intercept. In an exploratory analysis in the CBS group, we repeated this subject-level analysis while adding an additional main effect for global $^{18}$F-Flutemetamol amyloid-PET levels stratified at the median, to determine whether subthreshold levels of Aβ were associated with increased subcortical to cortical tau spread. Altering the definition of epicenters (i.e., 10, 15, 20%, 25, or 30 of ROIs with highest

18-F-PI-2620 PET binding) did not change the result pattern. All analyses were computed using R statistical software (r-project.org).

**Reporting summary**. Further information on research design is available in the Nature Research Reporting Summary linked to this article.

## Data availability

The fMRI data that are used in this study were obtained from the Alzheimer's disease Neuroimaging Initiative (ADNI) and are available from the ADNI database (adni.loni.usc.edu) upon registration and compliance with the data usage agreement. A list of ADNI RIDs that have been used for the current study can be found in the Supplementary Information file. ADNI neuroimaging data (unprocessed or processed) are available from the corresponding authors upon request and upon proof of approved access to the ADNI database. Neuroimaging data (i.e., unprocessed or processed PET & MRI images), as well as spreadsheets with postmortem and autoradiography data from PSP and CBS patients, have been used in previous publications[8, 27], and are available under restricted access from the corresponding author upon request and approval of a dedicated data usage agreement between institutions exchanging data. Data sharing of all data used in the current study is restricted since ethics approvals for PET and Postmortem studies or ADNI terms of use do not allow unrestricted and open-source sharing of patient-specific data with third-parties. Source data are provided with this paper.

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

## Acknowledgements

We acknowledge all members of the German Imaging Initiative for Tauopathies (GII4T) and the Alzheimer's Disease Neuroimaging Initiative (see supplementary information for a list of consortium members). We wish to thank the patients and their families, without whose support and altruism this research would not have been possible. Further, we would like to acknowledge Prof. Christian Haass (DZNE, Munich) for his consultation on study design, the European Reference Network for Rare Neurological Diseases—Project ID No 739510 and the Neurological Tissue Bank of the Biobanc-Hospital Clinic-IDIBAPS for sample and data procurement. This work was supported by the NIH (P30AG010124, P01AG066597, U19AG062418, and R01-NS109260), the National Center for Advancing Translational Sciences (TL1TR001880), the German Research Foundation (DFG; SCHR 774/5-1, DFG, INST 409/193-1 FUGG) the SyNergy excellence cluster (EXC 2145/ID 390857198). N.F. was supported by the Hertie foundation for clinical neurosciences. G.G.K. is supported by the Rossy Foundation and Safra Foundation. G.H. was funded by the Deutsche Forschungsgemeinschaft (DFG, HO2402/18-1 MSAomics), the German Federal Ministry of Education and Research (BMBF, 01EK1605A HitTau), VolkswagenStiftung and Lower Saxony Ministry for Science (Niedersächsisches Vorab), Petermax-Müller Foundation (Etiology and Therapy of Synucleinopathies and Tauopathies). M.B. was supported by the Hirnliga e.V. (Manfred-Strohscheer-Stiftung) and by the Alzheimer Forschung Initiative e.V. (grant number #19063p).

## Author contributions

N.F., M.B.: study concept and design, data analyses, interpretation of the results, drafting the manuscript; L.B., L.S., G.K., T.A., C.K., G.R., M.L., D.B., A.R., L.F., S.H., A.M., A.F., C.P., E.J., E.W., S.K., M.S., G.B., M.K., M.S., B.R., R.P., M.R., M.P., A.S., H.B., O.S., J.R., M.S., J.C., V.V., J.S., A.S., E.L., D.C., A.G., M.G., C.M., E.G., L.M.P., Y.C., J.S., L.D.L., C.T., S.A.S., J.R., S.X., D.I., S.R., J.H., M.S., P.B., V.L., J.T.: data acquisition, critical revision of the manuscript; J.L., G.H., M.E.: study concept and design, interpretation of the results, critical revision of the manuscript.

## Funding

## Competing interests

The authors declare no competing interests.

## Additional information

¹Institute for Stroke and Dementia Research, University Hospital of Munich, LMU Munich, Munich, Germany. ²Department of Nuclear Medicine, University Hospital of Munich, LMU Munich, Munich, Germany. ³Munich Cluster for Systems Neurology (SyNergy), Munich, Germany. ⁴Center for Neurodegenerative Disease Research (CNDR), Institute On Aging and Department of Pathology & Laboratory Medicine, University of Pennsylvania, Philadelphia, PA, USA. ⁵Tanz Centre for Research in Neurodegenerative Disease (CRND) and Department of Laboratory Medicine and Pathobiology, University of Toronto, Toronto, ON, Canada. ⁶Laboratory Medicine Program and Krembil Brain Institute, University Health

Network, Toronto, ON, Canada. [7]German Center for Neurodegenerative Diseases (DZNE), Munich, Germany. [8]Department of Psychiatry and Psychotherapy, University Hospital, LMU Munich, Munich, Germany. [9]Center for Neuropathology and Prion Research, LMU Munich, Munich, Germany. [10]Department of Neurology, Klinikum Rechts der Isar, Technical University of Munich, Munich, Germany. [11]Department of Neurology, Hannover Medical School, Carl-Neuberg-Str. 1, 30625 Hannover, Germany. [12]Clinic of Neurology, CCS, University of Belgrade, Belgrade, Republic of Serbia. [13]Life Molecular Imaging GmbH, Berlin, Germany. [14]Department of Neurology, University Hospital of Munich, LMU Munich, Munich, Germany. [15]Department of Radiology, University Hospital of Munich, LMU Munich, Munich, Germany. [16]Ageing Epidemiology Research Unit (AGE), School of Public Health, Imperial College, London, UK. [17]Department of Nuclear Medicine, University Hospital Leipzig, Leipzig, Germany. [18]Department of Nuclear Medicine, University Hospital Cologne, Cologne, Germany. [19]Department of Neurology, University Hospital Leipzig, Leipzig, Germany. [20]Department of Molecular Imaging & Therapy, Austin Health, Heidelberg, VIC, Australia. [21]Department of Psychiatry, University of Pittsburgh, Pittsburgh, PA, USA. [22]Department of Medicine, Austin Health, The University of Melbourne, Melbourne, VIC, Australia. [23]InviCRO, LLC, Boston, MA, USA. [24]Molecular Neuroimaging, A Division of inviCRO, New Haven, CT, USA. [25]Frontotemporal Degeneration Center, University of Pennsylvania, Philadelphia, PA, USA. [26]Department of Neurosciences, University of California, La Jolla, San Diego, CA, USA. [27]Department of Neurology, University of Pennsylvania, Philadelphia, PA, USA. [28]Neurological Tissue Bank and Neurology Department, Hospital Clínic de Barcelona, Universitat de Barcelona, IDIBAPS, CERCA, Barcelona, Catalonia, Spain. [29]Institute of Neurology, Medical University of Vienna, Vienna, Austria. [30]Parkinson's Disease & Movement Disorders Unit, Hospital Clínic / IDIBAPS / CIBERNED (CB06/05/0018-ISCIII), / European Reference Network for Rare Neurological Diseases (ERN-RND) / Institut de Neurociències (Maria de Maeztu Center), Universitat de Barcelona, Barcelona, Catalonia, Spain. [31]Department of Neurology, Erasmus Medical Centre, Rotterdam, The Netherlands. [32]Department Clinical Genetics, Erasmus Medical Center, Rotterdam, The Netherlands. [33]London Neurodegenerative Diseases Brain Bank, Institute of Psychiatry, Psychology and Neuroscience, Kings College London, London, UK. [34]Department of Biostatistics, Epidemiology and Informatics, University of Pennsylvania, Philadelphia, PA, USA. [35]Department of Neurology, Hannover Medical School, Hannover, Germany. [36]These authors contributed equally: Nicolai Franzmeier, Matthias Brendel. [37]These authors jointly supervised this work: Günter Höglinger, Michael Ewers. ✉email: Nicolai.franzmeier@med.uni-muenchen.de

