## [Peer Review File · Nature Communications]

Tau deposition patterns are associated with functional connectivity in primary tauopathiesREVIEWER COMMENTS

Reviewer #1 (Remarks to the Author):

Since the development of first-generation tau tracers, the examination of tau spread via structural or functional connectivity has grown and provided important new insights into Alzheimer's disease pathophysiology and has largely verified autopsy reports. These tracers were sensitive to 3/4 repeat tau and had less affinity for diseases associated with 4 repeat tau. The second-generation tau tracers, in particular 18F-PI-2620 provide novel opportunities to examine similar disease mechanisms in 4 repeat tau neurodegenerative disorders, such as cortico-basal degeneration (CBS) and progressive supranuclear palsy (PSP). This is the goal of this work. This manuscript is very well-written, I enjoyed reading it. The methods have been validated in previous reports. The sample size is small, but these patients are rare and hard to find. Applying these methods and tackling these research questions in these patients is novel. In addition, the authors also prevented possible biases by excluding CBS patients with elevated amyloid pathology. The combination of in vivo with postmortem data is unique, strengthens the in vivo findings and provides new important insights for the field: inter-regional connectivity was associated with inter-regional tau burden, in particular for subcortical regions. Similar findings were observed for the postmortem data for PSP patients. In addition, the postmortem data demonstrated that tau covariance may be specific to neurons and oligodendroglia cells and not astroglia cells. This is an important novel finding.

I have several follow-up comments:

1. How would atrophy play a role in these associations? I noticed that the authors did not adjust for volume/morphology or did not seem to apply partial volume correction on the tau data?
2. Figure 1A: the brains are small, but it seems that controls also have relatively high binding in the subcortical regions (comparable to the patient groups). To my understanding, that is not expected, and I wonder if this may have biased the comparisons with the patient groups. How do the authors interpret this? Is this off-target binding?
3. The tau data of the postmortem cases was measured in an ordinal scale. Was a specific scale used to determine the ordinal nature of the tau rating? Was a count (e.g. number of tangles per mm²) available? It seemed that the authors used parametric linear statistical analyses (Pearson correlation or linear regression) to covary tau data? Given the nature of the ratings, would the results change if the authors applied ordinal methods here?
4. Figure 5: In these individual patterns, not everyone seems to follow the expected pattern, where in some cases Q3 has higher tau-connectivity associations as for Q1, in particular in the PSP. Is there something about these cases that explains their variable pattern?
5. Do the authors have clinical (behavior/cognitive) data to link this back to these connectivity-tau patterns? It would be interesting to see if there is specific pattern to motor or cognitive functions.
6. All data in this manuscript is cross-sectional, and the analyses are correlational and as such it is difficult to determine that this is truly "spreading" of tau. Therefore, I would advise the authors to limit the use of the word spreading in the title, abstract and discussion.

Excellent work!
Heidi Jacobs

Reviewer #2 (Remarks to the Author):

The authors of this report have continued a research theme in which they examine how the pattern of tau deposition is related to the pattern of functional connectivity, inferring that tau spread is facilitated by trans-neuronal transport of the pathological protein. In this case, the authors used data from 46 patients with either Corticobasal Syndrome (CBS) or Progressive Supranuclear Palsy (Richardson's Syndrome; PSP-RS) with the tau PET tracer PI-2620 in conjunction with PET data from 15 cognitively normal (CN) controls and functional connectivity data from 69 amyloid- and tau-negative ADNI subjects. In short, the authors found a high correlation between inter-regional patterns of functional

connectivity and patterns of tau deposition, and also that tau epicenters (defined as regions with high tau deposition) were highly correlated with tau deposition in regions with which they were functionally connected. In CBS patients, subthreshold but higher beta-amyloid levels were associated with higher tau deposition in cortex. In two separate postmortem samples of PSP patients examined for tau pathology, the authors found that regional correlations of this pathology in a new set of regions (defined by the pathological examination) paralleled the functional connectivity as well.

This is an interesting study which, while following on a series of observations using similar methods to study Alzheimer's disease, is novel in showing similar effects of tau spread in non-AD tauopathies with a different PET tracer. The methods are elegant and the work is clearly described and illustrated. Despite these strengths I think that there are a number of substantial problems with the work.

1. The tracer, PI-2620, as the authors readily admit, has not been validated through autopsy confirmation to bind to 4R tau found in these diseases. To my knowledge, this research group is in fact the only group to report that it binds to 4R tau in vivo using PET (Brendel et al, JAMA Neurology 2020), when other tau PET tracers developed for the 3R/4R tau found in Alzheimer's disease (as this one was) do not bind to 4R tau. The existing evidence is complicated; another study (Tezuka et al, Brain Communications) did not find evidence of binding of PI-2620 to 4R tau and no correlation between in vivo and postmortem binding. However that group used a different imaging time window. Brendel et al, also did not find a relationship between PI-2620 binding and disease severity. All in all, the evidence for PI-2620 as a validated ligand for 4R tauopathies must be described as tentative at this point.

2. I am also concerned about a major technical issue in data acquisition and analysis. Multiple PET scanners were used and their resolutions were brought to a common value of 9 mm x 9 mm x 10 mm. Yet the study was primarily focused on the brain regions with the highest tracer deposition in subcortical structures. These structures cannot be quantitated as separate with this resolution. I understand that the authors included a covariate for Euclidean distance, but I cannot see how this could solve the problem that spatially proximate regions have substantial spill-in and spill out of tracer, leading to spurious correlations. I don't know what the authors could do to convince me but I think at minimum they need to examine a partial volume correction and also show the data both with and without the distance correction to see if there is a substantial effect from this statistical correction.

3. The cross-sectional nature of the study is limiting in terms of being able to ascribe causality to these associations. Use of words like "spreading" and "progression" are not supported by cross sectional data but require longitudinal measurement.

4. The PET control subjects in the study were younger in age than the PSP-RS patients and no information about age is provided for the ADNI participants who contributed the functional connectivity data. The authors should note this as a limitation and indicate how it could have affected the results.

5. The authors acknowledge that the use of a separate group for postmortem validation is a limitation. This is understandable. However, to show that the autopsy data parallel the in vivo data they can return to their PET sample and examine the correlations between ROIs in the PET and functional connectivity data in the same ROIs as used with the autopsy data to see if these same regions that demonstrate correlation in the autopsy data also demonstrate similar correlations in the PET data.

6. The amyloid effects in the CBS participants are interesting and compelling. However, to show the specificity for this disease, the authors should also examine a similar relationship in the PSP-RS cases.

Reviewer #3 (Remarks to the Author):

This study investigated whether connectivity drives 4R-tau spreading patterns by combining resting-state fMRI connectomics with both 18F-PI-2620 tau PET in 46 patients with clinically diagnosed 4RTs and postmortem cell-type-specific regional assessments from two independent PSP samples. The authors claimed that patient-level tau patterns can be predicted by the connectivity of subcortical tau epicenters. The topic dealt with in this study is clinically important and the data is clearly presented and well organized. I have a few concerns in this manuscript before being published in this journal.

Major concerns

1. The authors used connectivity templates from healthy controls for combining with PSP/CBS patients' data. Given that PSP/CBS patients showed different connectivity patterns in some previous studies, this seems to be a concern. The authors should make further discussion on this in the manuscript. To convince readers about this issue, using another 4R tau ligand, such as 18F-PM-PBB3 (18F-APN 1607), using the same analysis technique would be another option in the future study. If the results were similar in two ligands, the claim in this manuscript would be more convincing. The authors can discuss about this point by referring to the paper (Neuron 2021 Jan 6;109(1):42-58.e8).

2. As was discussed about the meaning of rsfMRI in the manuscript (467-477), the origin of rsfMRI signals has not been completely understood. Thus, it may not be completely reasonable to use rsfMRI data to investigate the neuronal connectivity in this study. Some studies claimed that the origin is derived from other components such as glia. Given that PSP/CBS patients have abundant tau in astrocytes, this issue, about the origin of rsfMRI signals, should be discussed more carefully. Also, regarding the title of this manuscript, the words "neuronal connectivity" might not be appropriate. If the authors would like to use these words, DTI data would be preferable by combining with tau PET data, which would be possible by acquiring from ADNI. Otherwise more comprehensive discussion would be necessary to claim that rsfMRI represents neuronal connectivity.

Minor concerns

1. I think there is a possibility that there are left-right differences in the accumulation of CBS, but did the analysis take these differences into consideration?

2. For tau PET, there are likely to be differences among facilities and machines, but how did you ensure these differences?

3. Compared with probable CBS & PSP, can you say that the trend of possible CBS & PSP is similar to the present results?

4. In the CBS>controls group comparison, there seems to be a predominance of accumulation in the occipital lobe and tip of frontal lobe. Are these results consistent with previously reported pathological findings?

5. What does the accumulation around the red nucleus in the control group indicate?

6. In the introduction (line 108), authors described that PSP and CBD share core genetic, biochemical and neuropathologic features. However, PSP and CBD show different biochemical properties and can be distinguished by immunoblotting of sarkosyl-insoluble tau (Arai et al., Ann Neurol 2004, Taniguchi-Watanabe et al., Acta Neuropathol 2016). Furthermore, cryo-EM analyses of tau filaments prepared from the diseased brains diagnosed neuropathologically elucidated the folding structures of tau in various tauopathies including CBD and PSP and confirmed the biochemical differences, and structure-based classification of tauopathies are proposed in Nature this week (Shi et al., Nature 2021). Authors should cite these papers and revise their description.

Reviewer #1 (Remarks to the Author):

Since the development of first-generation tau tracers, the examination of tau spread via structural or functional connectivity has grown and provided important new insights into Alzheimer's disease pathophysiology and has largely verified autopsy reports. These tracers were sensitive to 3/4 repeat tau and had less affinity for diseases associated with 4 repeat tau. The second-generation tau tracers, in particular 18F-PI-2620 provide novel opportunities to examine similar disease mechanisms in 4 repeat tau neurodegenerative disorders, such as cortico-basal degeneration (CBS) and progressive supranuclear palsy (PSP). This is the goal of this work. This manuscript is very well-written, I enjoyed reading it. The methods have been validated in previous reports. The sample size is small, but these patients are rare and hard to find. Applying these methods and tackling these research questions in these patients is novel. In addition, the authors also prevented possible biases by excluding CBS patients with elevated amyloid pathology. The combination of in vivo with postmortem data is unique, strengthens the in vivo findings and provides new important insights for the field: inter-regional connectivity was associated with inter-regional tau burden, in particular for subcortical regions. Similar findings were observed for the postmortem data for PSP patients. In addition, the postmortem data demonstrated that tau covariance may be specific to neurons and oligodendroglia cells and not astroglia cells. This is an important novel finding.

We'd like to thank the reviewer for these encouraging remarks!

I have several follow-up comments:

1. Reviewer: How would atrophy play a role in these associations? I noticed that the authors did not adjust for volume/morphology or did not seem to apply partial volume correction on the tau data?

Response: This is an important comment. We agree that regional grey matter atrophy may indeed lead to over- or underestimation of tau-PET SUVRs across different brain regions in patients with 4R tauopathies.¹ As suggested by the reviewer, this may be addressed by applying partial volume effect (PVE) correction to adjust the tau-PET signal for differences in grey matter volume. To this end, we used the Geometric Transfer Matrix (GTM) method² to determine PVE corrected tau-PET SUVRs for the 32 subcortical ROIs (i.e. TIAN atlas) and the 200 cortical ROIs (i.e. Schaefer atlas). In order to assess the effect of PVE-correction on our findings, we reran all ROI-based tau-PET analyses using PVE corrected data. Here, all results remained consistent with those presented in the initially submitted version of manuscript, as summarized in supplementary Figures 1-3. Specifically, we found a strong correlation between functional connectivity and covariance in tau-PET for subcortical (PSP-RS: $\beta=0.567$, $p<0.001$; CBS: $\beta=0.524$, $p<0.001$, Supplementary Figure 1A&B) and cortical regions (PSP-RS: $\beta=0.428$, $p<0.001$; CBS: $\beta=0.379$, $p<0.001$, Supplementary Figure 1C&D). Using group-level data, there was a strong association between connectivity of subcortical tau epicenters and tau-PET signals in connected subcortical (PSP-RS: $\beta=0.840$, $p<0.001$; CBS: $\beta=0.822$, $p<0.001$, Supplementary Figure 2A&B) and cortical regions (PSP-RS: $\beta=0.603$, $p<0.001$; CBS: $\beta=0.639$, $p<0.001$, Supplementary Figure 2G&H). For subject-level data, we also found a significant association between epicenter connectivity and tau-PET signals in Q1-Q4 ROIs for the subcortex (PSP-RS: $b/SE=0.092/0.015$, $p<0.001$; CBS: $b/SE=0.088/0.012$, Supplementary Figure 3B&D) and cortex (PSP-RS: $b/SE=0.032/0.008$, $p=0.008$; CBS: $b/SE=0.041/0.005$, Supplementary Figure 3C&E). These additional analyses suggest that our findings on connectivity vs. tau-PET patterns in 4R tauopathy patients are not specifically driven or biased by brain atrophy. As pointed out above, we have added these additional PVE-corrected analyses to the results and supplementary data of the manuscript and we have added a brief description of the PVE-correction to the methods section of the manuscript (p. 26).

2. Reviewer: Figure 1A: the brains are small, but it seems that controls also have relatively high binding in the subcortical regions (comparable to the patient groups). To my understanding, that is not expected, and I wonder if this may have biased the comparisons with the patient groups. How do the authors interpret this? Is this off-target binding?

Response: The reviewer is correct that there is considerable tracer retention in controls in brainstem nuclei, which was similarly reported in our previous paper on the evaluation of 18F-PI-2620 tau-PET in 4R tauopathy patients.³ Elevated 18F-PI-2620 signal in brainstem nuclei most likely reflects off-target binding to neuromelanin (e.g. substantia nigra), to which the 18F-PI-2620 tracer has been previously shown to bind.⁴ However, even with the strong underlying neuromelanin binding, 18F-PI-2620 binding in the substantia nigra of patients with PSP-RS was elevated when compared to controls.³ Medial-temporal lobe binding may reflect off-target binding in the choroid plexus and/or typical age-related tau deposition.⁵ Thus, we agree with the reviewer that off-target binding in

brainstem or plexus-near brain regions may bias voxel-wise group comparisons between PSP-RS/CBS patients and controls, hence drawing clear conclusions about normality/abnormality of tau-PET in these regions is difficult. However, severe off-target binding is not observed in typical PSP-RS or CBS target sites (e.g. pallidum, putamen, caudate)³ or cortical regions, suggesting that off-target binding does not systematically confound the ROI-based analyses of tau-PET vs. functional connectivity. Binding of PI-2620 to 4R tau pathology in cortical and subcortical target regions is further supported by our autoradiography vs. immunohistochemistry analyses, showing a correspondence between AT8 stained tau and PI-2620 binding in post mortem samples derived from PSP-RS patients (see our response to comment 1 by reviewer 2 and Figure 2 in the manuscript). Still, we are aware that 18F-PI-2620 is not free from off-target binding,^{3, 4} which we now briefly discuss in the limitations section of the manuscript (p. 21).

3. Reviewer: The tau data of the postmortem cases was measured in an ordinal scale. Was a specific scale used to determine the ordinal nature of the tau rating? Was a count (e.g. number of tangles per mm²) available? It seemed that the authors used parametric linear statistical analyses (Pearson correlation or linear regression) to covary tau data? Given the nature of the ratings, would the results change if the authors applied ordinal methods here?

Response: This is an excellent suggestion. It is correct that all post-mortem tau data have been rated on an ordinal scale based on expert ratings (i.e. none, mild, moderate, severe) by trained neuropathologists, as described previously for the post-mortem PSP datasets.⁶ While we appreciate that full quantitative data (e.g. tangle count per mm²) would be of high value, those data have not been obtained in the current datasets. Since the currently used data is on an ordinal scale, we agree that a non-parametric approach such as Spearman correlation is potentially better suited to assess covariance in tau. Thus, we reran the analyses on covariance in post-mortem tau levels, this time using partial Spearman correlation, while accounting for age at death and sex as implemented in the *pcor* function of R-package *ppcor* (<https://cran.r-project.org/web/packages/ppcor/ppcor.pdf>). When using this alternative non-parametric approach to determine the inter-regional covariance in post-mortem tau levels, all results remained fully consistent with the analyses in the initially submitted version of the manuscript. Specifically, we found a strong correlation between covariance in post-mortem tau and functional connectivity both in the Munich ($\beta=0.503$, $p<0.001$, Fig.6G) and UPENN dataset ($\beta=0.539$, $p<0.001$, Fig.6H), controlling for Euclidean distance between the ROIs. Similarly, the strongest association between connectivity and covariance in tau was found for neuronal tau, followed by oligodendroglial and astrocytic tau (see Figs.6I&J). Together, these results suggest that usage of a non-parametric approach yields consistent results with those presented in the manuscript. We have replaced the original analyses using Pearson correlation with these new analyses using Spearman correlation.

4. Reviewer: Figure 5: In these individual patterns, not everyone seems to follow the expected pattern, where in some cases Q3 has higher tau-connectivity associations as for Q1, in particular in the PSP. Is there something about these cases that explains their variable pattern?

Response: This is indeed an interesting observation. A potential explanation is that our tau-PET approach using a subcortical brain atlas⁷ does not capture all relevant epicenters with highest tau pathology, especially not those falling in brainstem nuclei, which are among the earliest sites of tau accumulation in PSP⁶. Thus, it is possible that connectivity-based spreading from tau-harboring brainstem nuclei may contribute to subcortical tau accumulation which does not necessarily follow the connectivity pattern of those tau-PET-based epicenters based on which the topology of Q1-Q4 ROIs has been determined. However, tau-PET in brainstem nuclei is complicated by off target binding in these regions^{3, 4} (see also our response on comment 1). Even further, determining functional connectivity of brainstem nuclei is technically highly demanding and requires correction for physiological artifacts (e.g. breathing, heart rate)⁸ and potentially a higher spatial resolution of fMRI data, which are not available from the ADNI database. Thus, additional analyses including also brainstem nuclei as potential tau epicenters could not be performed.

An alternative explanation could be related to the usage of a connectivity template that was derived from healthy control subjects, which does not capture disease-related connectivity changes. It is possible that PSP-RS patients show tau-related disruptions in subcortical connectivity,⁹ which may alter epicenter connectivity and thus change downstream tau accumulation patterns. To address this, it will be important for future studies to combine high-resolution subject-level connectivity data with tau-PET, in order to test whether individual connectivity patterns shape tau spreading in patients with 4R tauopathies. We have now briefly addressed this in the discussion section of the manuscript (p.21-22).

5. Reviewer: Do the authors have clinical (behavior/cognitive) data to link this back to these connectivity-tau patterns? It would be interesting to see if there is specific pattern to motor or

cognitive functions.

Response: This is an excellent point and it will be clinically highly relevant to map patterns of pathological brain changes to the cognitive and motor symptoms in patients with 4R tauopathies, as has been previously done for Alzheimer's disease patients.¹⁰ However, the currently included patients were compiled from multiple sites and although the PSP rating scale was available in most of the PSP cases we did not deem the inter-rater standardization across centers high enough to rely on single item severity. Furthermore, cognition was only measured by a screening test (e.g. MoCA) in most cases (i.e. available for 11/22 PSP-RS cases and 22/24 CBS cases). Please note that the PSP rating scale as well as the MoCA and the Schwab and England Activities of Daily Living scale (SEADL) have been added to table 1. In addition, we would like to point the reviewer to our previous publication, which did not find a clear association between the level of tau-PET and relatively gross measures of clinical disease severity or disease duration in PSP patients.³ Thus, clinical phenotypes in patients with 4R tauopathies may be more strongly determined by downstream consequences of tau pathology (e.g. neurodegeneration, glucose hypometabolism).¹¹ Still, we believe that there is a strong need for larger studies that perform detailed phenotyping of cognitive and motor symptoms in patients with 4R tauopathies, in order to develop a better understanding of how regional brain changes (e.g. tau, neurodegeneration, hypometabolism) relate to clinical disease manifestation. Thus, we have added a statement to the discussion of the manuscript, stating that "*The current results may be used as a starting point for future studies with detailed phenotyping of cognitive and motor function to map tau spreading patterns and downstream neurodegeneration to cognitive and motor phenotypes in patients with 4R tauopathies*" (p.22).

6. Reviewer: All data in this manuscript is cross-sectional, and the analyses are correlational and as such it is difficult to determine that this is truly "spreading" of tau. Therefore, I would advise the authors to limit the use of the word spreading in the title, abstract and discussion.

Response: We appreciate the reviewers' concern and have rephrased the according sections. In addition, we have changed the title to "*Tau deposition patterns are associated with functional connectivity in primary tauopathies – evidence from tau-PET and histopathology*".

Excellent work!
Heidi Jacobs

Thanks a lot Dr. Jacobs for your encouraging and very helpful comments!

Reviewer #2 (Remarks to the Author):

The authors of this report have continued a research theme in which they examine how the pattern of tau deposition is related to the pattern of functional connectivity, inferring that tau spread is facilitated by trans-neuronal transport of the pathological protein. In this case, the authors used data from 46 patients with either Corticobasal Syndrome (CBS) or Progressive Supranuclear Palsy (Richardson's Syndrome; PSP-RS) with the tau PET tracer PI-2620 in conjunction with PET data from 15 cognitively normal (CN) controls and functional connectivity data from 69 amyloid- and tau-negative ADNI subjects. In short, the authors found a high correlation between inter-regional patterns of functional connectivity and patterns of tau deposition, and also that tau epicenters (defined as regions with high tau deposition) were highly correlated with tau deposition in regions with which they were functionally connected. In CBS patients, subthreshold but higher beta-amyloid levels were associated with higher tau deposition in cortex. In two separate postmortem samples of PSP patients examined for tau pathology, the authors found that regional correlations of this pathology in a new set of regions (defined by the pathological examination) paralleled the functional connectivity as well.

This is an interesting study which, while following on a series of observations using similar methods to study Alzheimer's disease, is novel in showing similar effects of tau spread in non-AD tauopathies with a different PET tracer. The methods are elegant and the work is clearly described and illustrated. Despite these strengths I think that there are a number of substantial problems with the work.

1. Reviewer: The tracer, PI-2620, as the authors readily admit, has not been validated through autopsy confirmation to bind to 4R tau found in these diseases. To my knowledge, this research group is in fact the only group to report that it binds to 4R tau in vivo using PET (Brendel et al, JAMA Neurology 2020), when other tau PET tracers developed for the 3R/4R tau found in Alzheimer's disease (as this one was) do not bind to 4R tau. The existing evidence is complicated; another study (Tezuka et al, Brain Communications) did not find evidence of binding of PI-2620 to 4R tau and no correlation between in vivo and postmortem binding. However, that group used a different imaging time window. Brendel et al, also did not find a relationship between PI-2620 binding and disease severity. All in all, the evidence for PI-2620 as a validated ligand for 4R tauopathies must be described as tentative at this point.

Response: We agree with the reviewer that large-scale autopsy validation studies of next generation tau-PET tracers, including [¹⁸F]PI-2620, in 4R-tauopathies are still missing.

However, even autopsy validation has several caveats, such as time gaps between PET and autopsy as well as variable tissue fixation times. Furthermore, full kinetic PET quantification can rarely be performed within these studies¹² since end-of-life patients do not well tolerate long lasting scans and arterial sampling is ethically questionable in severely impaired patients. Thus, other indicators beyond direct autopsy validation should be considered as well to adequately judge on potential binding of next generation tau PET tracers to 4R tau:

- Our past studies administering Pi2620 in 4R-tauopathies (Brendel et al., 2020; Palleis et al., 2021; Song et al., 2021) relied on clinical diagnostic criteria which are validated against the autopsy-standard and therein demonstrated excellent specificity to detect 4RT-pathology in general, and PSP-pathology specifically (PMID: 32914550; 31571273).
- Furthermore, a recent study refined binding sites from cryo-EM by metadynamics simulations in order to provide atomic resolution of the binding modes and thermodynamic properties.¹³ Several binding sites for PI-2620 were observed on 4R Tau CBD fibrils which support sufficient PI-2620 in vivo binding to 4R tau.
- In addition, we have performed several novel analyses in order to assess the capability of PI-2620 to quantify 4R tau pathology. First, we determined the affinity of PI-2620 vs. Flortaucipir using an in-vitro competition assay with self-blocking. Here, we found a much lower IC50 for PI-2620 (2.7nM) vs. Flortaucipir (18.4nM), suggesting stronger binding of PI-2620 to 4R tau fibrils compared to Flortaucipir (Fig.2A). Similar data have been obtained by Mathis, Klunk and Ikonomic, as presented at the Tau 2020 Meeting (K_D PSP-tissue: 3H-PI-2620: 8.1 nM, 3H-Flortaucipir 12 nM), suggesting stronger binding of PI-2620 to 4R tau than Flortaucipir. However, earlier data also indicated a certain binding affinity of Flortaucipir to 4R tau including a correlation with autopsy-assessed tau levels in single cases,^{14, 15} yet this was not reproducible in a larger sample.¹² Despite a potential affinity of Flortaucipir to 4R tau, massive off-target binding of Flortaucipir in basal-ganglia drastically complicates the assessment of specific binding to a potentially underlying tau pathology in PSP. In contrast,

PI-2620 shows only little basal ganglia off-target binding in controls, suggesting a critical advantage of PI-2620 to quantify subcortical 4R tau pathology.³

- To further assess the capability of PI-2620 to bind to 4R tau pathology, we performed combined PI-2620 autoradiography and AT8 immunohistochemistry in 233 post-mortem probes derived from 16 histopathologically confirmed PSP-RS cases across three subcortical and cortical brain regions (i.e. frontal cortex: n=105, pallidum: n=56; putamen: n=72). Within each sample, AT8 binding levels were assessed semiquantitatively by expert neuropathologists (i.e. low=+, medium=++, high=+++), and the autoradiography signal was quantified as the ratio of PI-2620 signal in the target tissue (i.e. frontal cortex, pallidum, putamen) divided by PI-2620 signal in AT8 negative white matter. Using Spearman correlation, we found a positive association between semiquantitative AT8 assessments and PI-2620 autoradiographic signal (Frontal cortex: $r=0.44$, $p<0.001$; Pallidum: $r=0.47$, $p<0.001$; Putamen: $r=0.4$, $p<0.001$), as well as significant differences in PI-2620 autoradiography signals between semiquantitatively assessed AT8 binding groups (i.e. low, medium, high) using ANOVAs (Frontal cortex: $F=10.42$, $p<0.001$; Putamen: $F=12.08$, $p<0.001$; Pallidum: $F=8.36$, $p<0.001$, Fig.2B-D). Together, these novel results suggest that PI-2620 is potentially capable of detecting 4R tau pathology. Supporting this, our previous work showed that PI-2620 signal matched typical 4R tau distribution patterns in PSP³ and CBS¹⁶ when using dynamic scanning and earlier static windows¹⁷. However, no or low detectable signal is found in late static windows in 4R-tauopathies in the recent study by Tezuka et al,¹⁸ which may relate to faster tracer dissociation from 4R tau when compared to 3/4R tau.¹⁹ Less favorable characteristics for [¹⁸F]PI-2620 binding to 4R tau when compared to 3/4R tau in docking studies support this in vivo finding.¹³ In summary, there are several indicators that speak for a 4R tau related PI-2620 in vivo signal, which may be enriched by signal sources closely associated with tau pathology. The novel integrated data strengthen the hypothesis of 4R tau binding by PI-2620. However, we fully agree with the reviewer that we should be tentative unless larger autopsy cohorts, arterial sampling data and phase 2/3 trials are completed. As the reviewer noted, our spatially matched tau PET and immunohistochemistry investigation (see response on comment 5) is another important step to further interrogate and validate the in vivo PI-2620 PET signal in 4R-tauopathies. All additional analyses have been added to the results (p.8) and methods section (p.27-28). In addition, we briefly discuss the capability of PI-2620 to bind to 4R tau pathology in the discussion section (p.20-21).

2. Reviewer: I am also concerned about a major technical issue in data acquisition and analysis. Multiple PET scanners were used and their resolutions were brought to a common value of 9 mm x 9 mm x 10 mm. Yet the study was primarily focused on the brain regions with the highest tracer deposition in subcortical structures. These structures cannot be quantitated as separate with this resolution. I understand that the authors included a covariate for Euclidean distance, but I cannot see how this could solve the problem that spatially proximate regions have substantial spill-in and spill out of tracer, leading to spurious correlations. I don't know what the authors could do to convince me but I think at minimum they need to examine a partial volume correction and also show the data both with and without the distance correction to see if there is a substantial effect from this statistical correction.

Response: The reviewer raises an important point, regarding the ability of PET to discriminate between spatially adjacent subcortical structures. The reviewer points out correctly that images were brought to a common resolution of 9x9x10mm, yet we would like to highlight that these data were exclusively used for voxel-wise group comparisons between patients and controls in MNI standard space (see Fig.1), in order to minimize the bias of scanning resolution for comparisons on the voxel level. This resolution was chosen based on the relatively low resolution of the PET scanner in New Haven, which was used for tau-PET assessment in several control subjects. In contrast, PET data from PSP-RS and CBS patients was recorded with a higher resolution (5x5x5mm for patients scanned in Munich, 3x3x3mm for patients scanned in Leipzig). Thus, SUVR extraction of ROI-based data for PSP-RS and CBS patients was performed in PET native space without additional smoothing in order to minimize spill-over between subcortical regions. Specifically, we warped patient-specific grey matter masks from T1-MRI tissue segments as well as the cortical and subcortical atlases to PET space in order to define cortical grey matter and subcortical regions directly on PET images. Thus, subject-level ROI extraction is not additionally confounded by image smoothing to a uniform resolution of 9x9x10mm. To further minimize the bias introduced by different scanning parameters, we statistically accounted for different scanning protocols by i) partialling out the effect of imaging protocols while determining the covariance in inter-regional tau-PET and ii) using site as a covariate for all subject-level analyses. In addition, controlling for Euclidean distance should partially account for higher tau covariance among ROIs in close physical proximity. However, we agree with the reviewer that additional analyses using partial volume effect (PVE)-corrected data may further help

to assess whether our results are systematically influenced by different scanner resolution. To this end, we applied PVE-correction using the Geometric transfer matrix (GTM) method, which is particularly well-suited to correct ROI-based PET data. As pointed out already in our response to comment 1 by reviewer 1, all results obtained with PVE-corrected data remained fully consistent with the analyses presented in the initially submitted version of the manuscript. All analyses using PVE-corrected data are now illustrated in Supplementary Figs.1-3, and a more detailed description of these analyses can be found in our response to comment 1 by reviewer 1. In addition, the reviewer suggested to rerun our statistical models without controlling for Euclidean distance, to determine whether this statistical correction has a substantial effect on our results. When omitting Euclidean distance as a covariate, we find consistent results with those presented in the main manuscript, both when using PVE-corrected and non-PVE-corrected data. These additional analyses are now summarized in supplementary tables 1&2 for non-PVE corrected data and supplementary tables 3&4 for PVE-corrected data and are mentioned in the results section of the manuscript. Together, these additional analyses support the view that our results are not systematically biased by partial volume effects or by correcting for distance between ROIs.

3. Reviewer: The cross-sectional nature of the study is limiting in terms of being able to ascribe causality to these associations. Use of words like “spreading” and “progression” are not supported by cross sectional data but require longitudinal measurement.

Response: We appreciate this concern, which was also raised by reviewer 1, and we are certainly aware that our cross-sectional study design does not allow to make any causal conclusions on the link between connectivity and the spreading of tau. Thus, we have rephrased sections of the manuscript which may have implied a causal association. In addition, we have changed the study title to *“Tau deposition patterns are associated with functional connectivity in primary tauopathies – evidence from tau-PET and histopathology”*, in order to avoid any misunderstanding.

4. Reviewer: The PET control subjects in the study were younger in age than the PSP-RS patients and no information about age is provided for the ADNI participants who contributed the functional connectivity data. The authors should note this as a limitation and indicate how it could have affected the results.

Response: The reviewer correctly notes that control subjects were younger than PSP-RS patients, which may potentially bias voxel-wise group-comparisons in tau-PET uptake. However, we would like to mention that all voxel-wise group comparisons have been corrected for age, sex, and study site, hence age effects should be statistically accounted for. Nevertheless, we have added a short statement to the limitations section, stating that the age difference between PSP-RS patients and controls may to a certain degree affect voxel-wise group comparisons in tau-PET levels (p. 20). Of note, the control subjects are not included in any of the analyses on ROI-based tau-PET vs. functional connectivity, hence the younger age of control subjects does not confound our main analyses. In addition, we have added demographic information (i.e. age, sex) of the ADNI resting-state fMRI sample to the results section, which did not deviate statistically ($p > 0.05$) from the PSP-RS and CBS groups (p.7)

5. Reviewer: The authors acknowledge that the use of a separate group for postmortem validation is a limitation. This is understandable. However, to show that the autopsy data parallel the in vivo data they can return to their PET sample and examine the correlations between ROIs in the PET and functional connectivity data in the same ROIs as used with the autopsy data to see if these same regions that demonstrate correlation in the autopsy data also demonstrate similar correlations in the PET data.

Response: This is an excellent suggestion! Applying the same brain parcellation to tau-PET and post-mortem data is an elegant approach to replicate the association between functional connectivity and covariance in tau pathology across in vivo and histopathological (i.e. gold-standard) tau assessments. In addition, this analysis approach may also help further strengthen the claim that the tau-PET analyses indeed reflect covariance in 4R tau pathology. Thus, we followed the reviewers' suggestion and have obtained tau-PET SUVRs within those ROIs that were determined for the post-mortem parcellations of the Munich and UPENN datasets (Figs.6A&B). Specifically, we warped back the Post-mortem ROIs (both for the Munich and UPENN sample) from MNI space to PET native space and obtained ROI-based SUVRs for each PSP-RS patient. We then determined the covariance in tau-PET between the post-mortem ROI parcellations and tested whether higher connectivity is associated with higher covariance in tau-PET using the Munich-based or UPENN-based post-mortem parcellation (i.e. Figs.6A&B). Here, we found a strong association between functional connectivity and tau-PET using the Munich ($\beta = 0.66$, $p < 0.001$) or UPENN parcellation ($\beta = 0.52$, $p < 0.001$), controlling for Euclidean distance between ROIs. These results remained consistent without controlling for Euclidean

distance as a covariate (Munich: $\beta=0.43$, $p<0.001$, UPENN: $\beta=0.33$, $p=0.005$). Together, these additional analyses suggest that applying the same brain parcellation to post-mortem histopathological 4R tau data and in-vivo tau-PET assessments from PSP patients yields consistent associations between functional connectivity and covariance in tau pathology. We have added these additional analyses to the results section of the manuscript (p.14).

6. Reviewer: The amyloid effects in the CBS participants are interesting and compelling. However, to show the specificity for this disease, the authors should also examine a similar relationship in the PSP-RS cases.

Response: We agree with the reviewer that it would be interesting to test a potential association between subthreshold $A\beta$ levels and cortical tau accumulation also in PSP-RS patients. However, [^{18}F]flutemetamol amyloid-PET was only available for one out of 22 PSP-RS patients, as PET-based $A\beta$ -status was only systematically assessed in patients with CBS in which underlying AD pathology is a clinically relevant differential diagnosis. In PSP patients, CSF $A\beta$ -status was obtained as a binary variable, and continuous data are not available to the researchers. Thus, conducting the same analysis in PSP-RS is not possible with the currently available data. We have added a sentence to the discussion section of the manuscript, stating that the effect of subthreshold amyloid-levels on cortical tau spreading remains to be investigated in PSP-RS patients (p.20).

Reviewer #3 (Remarks to the Author):

This study investigated whether connectivity drives 4R-tau spreading patterns by combining resting-state fMRI connectomics with both 18F-PI-2620 tau PET in 46 patients with clinically diagnosed 4RTs and postmortem cell-type-specific regional assessments from two independent PSP samples. The authors claimed that patient-level tau patterns can be predicted by the connectivity of subcortical tau epicenters. The topic dealt with in this study is clinically important and the data is clearly presented and well organized. I have a few concerns in this manuscript before being published in this journal.

Major concerns

1. Reviewer: The authors used connectivity templates from healthy controls for combining with PSP/CBS patients' data. Given that PSP/CBS patients showed different connectivity patterns in some previous studies, this seems to be a concern. The authors should make further discussion on this in the manuscript. To convince readers about this issue, using another 4R tau ligand, such as 18F-PM-PBB3 (18F-APN 1607), using the same analysis technique would be another option in the future study. If the results were similar in two ligands, the claim in this manuscript would be more convincing. The authors can discuss about this point by referring to the paper (Neuron 2021 Jan 6;109(1):42-58.e8).

Response: This is an important point, since disease-related connectivity changes may influence connectivity-based spreading patterns of tau pathology in PSP-RS or CBS patients. Please see also our response to comment 4 by reviewer 1, in which we address a similar topic. We agree with the reviewer that combining tau-PET with fMRI data of the same patients may help address the role of disease-related connectivity changes in the spreading of tau pathology. We have now added a section to the discussion of the manuscript in which we briefly address this limitation (p.22). In addition, we fully agree that replication of our findings with a second tracer that has been shown to bind to 4R tau (e.g. 18F-APN 1607) will be an important step in assessing tau spreading mechanisms in 4R tauopathies, which we now briefly highlight in the discussion of the manuscript (p.21). However, we would also like to point the reviewer to our analyses combining fMRI data with post-mortem data from two independent PSP patient samples, which provide post-mortem validation of the result pattern that was observed in the PI-2620 PET analyses.

2. Reviewer: As was discussed about the meaning of rsfMRI in the manuscript (467-477), the origin of rsfMRI signals has not been completely understood. Thus, it may not be completely reasonable to use rsfMRI data to investigate the neuronal connectivity in this study. Some studies claimed that the origin is derived from other components such as glia. Given that PSP/CBS patients have abundant tau in astrocytes, this issue, about the origin of rsfMRI signals, should be discussed more carefully. Also, regarding the title of this manuscript, the words 'neuronal connectivity' might not be appropriate. If the authors would like to use these words, DTI data would be preferable by combining with tau PET data, which would be possible by acquiring from ADNI. Otherwise more comprehensive discussion would be necessary to claim that rsfMRI represents neuronal connectivity.

Response: The reviewer is correct that the fMRI signal is based on multiple cellular sources and does not strictly reflect a neuronal signal.^{20, 21} However, fMRI connectivity has been shown to correlate with electrophysiological (i.e. EEG-based) measures of connectivity and brain activity^{22, 23} and fMRI-based functional connectivity has been shown to be constrained by the anatomical connections in the brain.²⁴ Thus, fMRI based connectivity is considered a suitable proxy to assess how brain regions are interconnected.²⁵ Still, we agree with the reviewer that the term "neuronal connectivity" may be an overstatement, hence we have rephrased the title to "*Tau deposition patterns are associated with functional connectivity in primary tauopathies – evidence from tau-PET and histopathology*". In addition, we now discuss the potential multi-cellular origin of the fMRI signal more carefully in the discussion section of the manuscript (p.21).

Further, the reviewer suggests to use DTI data instead of functional MRI data, since DTI may better reflect axonal connections in the brain. While this might be an advantageous approach for cortico-cortical connections, DTI cannot map fiber connections between spatially adjacent subcortical structures so far, and mapping connections of subcortical nuclei to cortical regions is also challenging from a methodological point of view. Thus, the major aims of the current study could not be addressed using DTI data at contemporary quality.

Minor concerns

3. Reviewer: I think there is a possibility that there are left-right differences in the accumulation of CBS, but did the analysis take these differences into consideration?

Response: We agree that CBS patients often show left-right differences in their tau accumulation pattern, yet, we decided not to flip CBS brain images so their pathology patterns fall on a common side, since flipping the images would limit the natural spatial heterogeneity and spatial variance in tau-PET patterns of patients with CBS. However, a higher spatial heterogeneity in tau-PET patterns facilitates the estimation of inter-regional covariance in tau-PET patterns, which was a key aim of the current study.

4. Reviewer: For tau PET, there are likely to be differences among facilities and machines, but how did you ensure these differences?

Response: This is an important comment, since multi-site and multi-scanner data acquisition may potentially bias our analyses. Thus, we corrected for site and scanner protocols on different levels. For voxel-wise group comparisons shown in Figure 1, we smoothed all tau-PET data to match a common resolution (i.e. defined by the scanner with the lowest spatial resolution) and added study site as a covariate. For assessing covariance in inter-regional tau-PET, we determined partial correlations in the inter-regional tau-PET signal, adjusting for age, sex as well as imaging protocols. Further, all statistical analyses using subject-level data were additionally adjusted for study site as a covariate.

In addition, we performed partial-volume effect (PVE) corrected analyses using scanner-specific image resolution to determine tau-PET SUVRs that were adjusted for regional grey matter volume and scanner resolution. When rerunning all analyses using the PVE-corrected data, all results remained consistent. For an in-depth description of these additional analyses, please see our response on comment 1 by reviewer 1, as well as supplementary Figs. 1-3 and supplementary tables 3&4 which summarize these analyses.

5. Reviewer: Compared with probable CBS & PSP, can you say that the trend of possible CBS & PSP is similar to the present results?

Response: This is an interesting comment, however, we refrained from rerunning the analyses stratified by subgroups of probable or possible PSP/CBS, since a further splitting into subgroups would limit statistical power and also further complicate an already complex set of analyses. We will certainly follow up this important question once larger numbers of observations will become available in the future.

6. Reviewer: In the CBS>controls group comparison, there seems to be a predominance of accumulation in the occipital lobe and tip of frontal lobe. Are these results consistent with previously reported pathological findings?

A previous post-mortem study that determined regional tau pathology in a small number of CBD patients, found relatively high rates of frontal tau pathology in CBD as well as some levels of occipital tau pathology, with the overall conclusion that tau distribution is relatively heterogeneous across CBD patients.²⁶ This suggests that occipital and frontal tau pathology can indeed be observed in CBS. However, a systematic large-scale investigation of tau pathology distribution as has been recently done for PSP is still missing in CBD,⁶ hence spreading and deposition patterns in CBD remain to be determined. Still, our autoradiographic analyses showed PI-2620 binding in regions with higher 4R tau, supporting the validity of the tau-PET binding patterns observed in the current study. In addition, the pattern of statistically significant PI-2620 group differences between CBS patients and controls reported in Figure 1 may be further influenced by an asymmetry in tau deposition that may underlie the CBS-typical asymmetry in clinical presentation. Patients were not flipped according to the hemispheric dominance of CBS symptoms to retain maximum spatial heterogeneity in tau deposition, which is essential for assessing inter-regional covariance (see our response to comment 3 by reviewer 3). Thus, the spatial heterogeneity in tau deposition may limit the sensitivity of common statistical approaches to detect elevated PI-2620 binding, hence abnormality in tau-PET may be underestimated.

7. Reviewer: What does the accumulation around the red nucleus in the control group indicate?

Response: Elevated tau-PET signal in the brainstem in controls and patients potentially reflects off-target binding to neuromelanin in the red nucleus and substantia nigra. We now mention this off-target binding in the limitation section of the manuscript (p.21). Please see also our response on comment 2 by reviewer 1.

8. Reviewer: In the introduction (line 108), authors described that PSP and CBD share core genetic, biochemical and neuropathologic features. However, PSP and CBD show different biochemical

properties and can be distinguished by immunoblotting of sarkosyl-insoluble tau (Arai et al., Ann Neurol 2004, Taniguchi-Watanabe et al., Acta Neuropathol 2016). Furthermore, cryo-EM analyses of tau filaments prepared from the diseased brains diagnosed neuropathologically elucidated the folding structures of tau in various tauopathies including CBD and PSP and confirmed the biochemical differences, and structure-based classification of tauopathies are proposed in Nature this week (Shi et al., Nature 2021). Authors should cite these papers and revise their description.

Response: These are reasonable points, hence we have removed the section in the introduction, and instead added a sentence to the discussion, stating that "*Recent studies have emphasized that CBD and PSP tau show distinguishable molecular characteristics,^{27, 28, 29} hence it will be important in future studies to assess whether molecular differences in CBD and PSP tau modulate the spreading potential of tau pathology.*"

REFERENCES

1. Shi HC, *et al.* Gray matter atrophy in progressive supranuclear palsy: meta-analysis of voxel-based morphometry studies. *Neurol Sci* **34**, 1049-1055 (2013).
2. Rousset OG, Ma Y, Evans AC. Correction for partial volume effects in PET: principle and validation. *J Nucl Med* **39**, 904-911 (1998).
3. Brendel M, *et al.* Assessment of 18F-PI-2620 as a Biomarker in Progressive Supranuclear Palsy. *JAMA Neurol* **77**, 1408-1419 (2020).
4. Kroth H, *et al.* Discovery and preclinical characterization of [(18)F]PI-2620, a next-generation tau PET tracer for the assessment of tau pathology in Alzheimer's disease and other tauopathies. *European journal of nuclear medicine and molecular imaging* **46**, 2178-2189 (2019).
5. Crary JF, *et al.* Primary age-related tauopathy (PART): a common pathology associated with human aging. *Acta Neuropathol* **128**, 755-766 (2014).
6. Kovacs GG, *et al.* Distribution patterns of tau pathology in progressive supranuclear palsy. *Acta Neuropathol* **140**, 99-119 (2020).
7. Tian Y, Margulies DS, Breakspear M, Zalesky A. Topographic organization of the human subcortex unveiled with functional connectivity gradients. *Nature neuroscience* **23**, 1421-1432 (2020).
8. Bar KJ, *et al.* Functional connectivity and network analysis of midbrain and brainstem nuclei. *NeuroImage* **134**, 53-63 (2016).
9. Gardner RC, *et al.* Intrinsic connectivity network disruption in progressive supranuclear palsy. *Ann Neurol* **73**, 603-616 (2013).
10. Ossenkoppele R, *et al.* Tau PET patterns mirror clinical and neuroanatomical variability in Alzheimer's disease. *Brain : a journal of neurology* **139**, 1551-1567 (2016).
11. Amtage F, *et al.* Functional correlates of vertical gaze palsy and other ocular motor deficits in PSP: an FDG-PET study. *Parkinsonism Relat Disord* **20**, 898-906 (2014).
12. Josephs K, *et al.* Relationship between (18)F-flortaucipir uptake and histologic lesion types in 4-repeat tauopathies. *J Nucl Med*, (2021).
13. Zhou Y, Li J, Nordberg A, Agren H. Dissecting the Binding Profile of PET Tracers to Corticobasal Degeneration Tau Fibrils. *ACS Chem Neurosci* **12**, 3487-3496 (2021).
14. Josephs KA, *et al.* [18F]AV-1451 tau-PET uptake does correlate with quantitatively measured 4R-tau burden in autopsy-confirmed corticobasal degeneration. *Acta neuropathologica* **132**, 931-933 (2016).

15. McMillan CT, *et al.* Multimodal evaluation demonstrates in vivo (18)F-AV-1451 uptake in autopsy-confirmed corticobasal degeneration. *Acta neuropathologica* **132**, 935-937 (2016).
16. Palleis C, *et al.* Cortical [(18) F]PI-2620 Binding Differentiates Corticobasal Syndrome Subtypes. *Mov Disord* **36**, 2104-2115 (2021).
17. Song M, *et al.* Feasibility of short imaging protocols for [(18)F]PI-2620 tau-PET in progressive supranuclear palsy. *Eur J Nucl Med Mol Imaging* **48**, 3872-3885 (2021).
18. Tezuka T, *et al.* Evaluation of [(18)F]PI-2620, a second-generation selective tau tracer, for assessing four-repeat tauopathies. *Brain Commun* **3**, fcab190 (2021).
19. Song M, *et al.* Binding characteristics of [(18)F]PI-2620 distinguish the clinically predicted tau isoform in different tauopathies by PET. *J Cereb Blood Flow Metab*, 271678X211018904 (2021).
20. Hillman EM. Coupling mechanism and significance of the BOLD signal: a status report. *Annu Rev Neurosci* **37**, 161-181 (2014).
21. Lu H, Jaime S, Yang Y. Origins of the Resting-State Functional MRI Signal: Potential Limitations of the "Neurocentric" Model. *Front Neurosci* **13**, 1136 (2019).
22. Wirsich J, *et al.* The relationship between EEG and fMRI connectomes is reproducible across simultaneous EEG-fMRI studies from 1.5T to 7T. *Neuroimage* **231**, 117864 (2021).
23. Shi Z, *et al.* On the Relationship between MRI and Local Field Potential Measurements of Spatial and Temporal Variations in Functional Connectivity. *Sci Rep* **9**, 8871 (2019).
24. Honey CJ, *et al.* Predicting human resting-state functional connectivity from structural connectivity. *Proc Natl Acad Sci U S A* **106**, 2035-2040 (2009).
25. Ma Y, *et al.* Resting-state hemodynamics are spatiotemporally coupled to synchronized and symmetric neural activity in excitatory neurons. *Proc Natl Acad Sci U S A* **113**, E8463-E8471 (2016).
26. Forman MS, *et al.* Signature tau neuropathology in gray and white matter of corticobasal degeneration. *Am J Pathol* **160**, 2045-2053 (2002).
27. Shi Y, *et al.* Structure-based classification of tauopathies. *Nature* **598**, 359-363 (2021).
28. Arai T, *et al.* Identification of amino-terminally cleaved tau fragments that distinguish progressive supranuclear palsy from corticobasal degeneration. *Ann Neurol* **55**, 72-79 (2004).

29. Taniguchi-Watanabe S, *et al.* Biochemical classification of tauopathies by immunoblot, protein sequence and mass spectrometric analyses of sarkosyl-insoluble and trypsin-resistant tau. *Acta Neuropathol* **131**, 267-280 (2016).

REVIEWER COMMENTS

Reviewer #1 (Remarks to the Author):

The authors have done a wonderful job in working through these comments and showing the robustness of their results. I look forward to see this published.

Reviewer #2 (Remarks to the Author):

Thank you for a very thorough response to my review

Reviewer #3 (Remarks to the Author):

The authors responded well to the points raised in the peer reviewers, and I thought there was no more comments with the content of the revised version of the manuscript.